

# Combining analytical solutions of Boussinesq equation with the modified Kozeny-Carman equation for estimation of catchment-scale hydrogeological parameters

Man Gao [1,2], Xi Chen [1,2]*, and Jintao Liu [3,4]

[1] Institute of Surface-Earth System Science, Tianjin University, Tianjin, China

5   [2] Tianjin Key Laboratory of Earth Critical Zone Science and Sustainable Development in Bohai Rim, Tianjin University, Tianjin, China

[3] State Key Laboratory of Hydrology-Water Resources and Hydraulic Engineering, Hohai University, Nanjing, China

[4] Department of College of Hydrology and Water Resources, Hohai University, Nanjing, China

*Corresponding to*: Xi Chen (xi_chen@tju.edu.cn)



**Abstract**: Saturated hydraulic conductivity ($K$), drainable porosity ($f$), and effective aquifer
thickness ($D$) are essential hydrogeological parameters for hydrologic modelling and
predicting. Streamflow recession analysis using analytical solutions of Boussinesq
equation can yield estimated values for two of these three hydrogeological parameters
when one is known a priori. In this study, we improved the inverse method for parameters
estimation by combining the modified Kozeny–Carman equation with analytical solutions
of Boussinesq equation to express the three hydrogeological parameters ($K$, $f$, and $D$) in
relation to catchment characteristics and recession constants in a sloping aquifer. Here, the
three parameters can be estimated simultaneously from streamflow recession analysis.
Results of the estimated parameters are compared with the field measurements and the soil-
texture based estimations in four small experimental catchments. It shows that our
estimated values of these catchment-scale parameters can represent equivalent values in
the measured aquifer profiles/sites. In hilly areas, the slope aquifer takes a vital effect on
the estimates of $K$ and $f$. Neglecting the sloping effect can lead to overestimation of $K$ and
underestimation of $f$ in 1 ~ 2 orders of magnitude in the study catchments. However, even
in the hilly catchments, the estimated aquifer thickness $D$ is much greater than that from
measurements on hillslopes while it approaches riparian thickness, indicating that the
riparian zone takes a vital role on flow recession and the parameter estimations.





## 1 Introduction

Saturated hydraulic conductivity ($K$), drainable porosity ($f$), and aquifer thickness ($D$) are

basic physical attributes of catchment and essential parameters for physically based

hydrologic models as well as for the hydrologic analysis of ungauged basins (Ali et al.,

2014; Bogaart et al., 2016; Carrillo et al., 2011; Li et al., 2014; Vannier et al., 2016).

Extensive measurements of these parameters to represent their intrinsic variations in a

catchment are costly and time-consuming. Thus, indirect methods, based on other more

easily measurable soil properties (such as soil porosity or texture) and/or regular

hydrologic observation data (such as streamflow) are sometimes preferred.

The Boussinesq equation is a general differential equation for groundwater flow in

unconfined aquifers of hillslope areas. The analytical and numerical solutions of the

Boussinesq equation illustrates streamflow behaviors in relation to catchment properties

like hillslope gradients, shapes, and hydrogeologic parameters of the aquifer (Troch et al.,

2013). Such analytical solutions have been widely used for estimating catchment-scale

hydrogeological parameters since the parameter estimation only needs the regular

streamflow observations (Brutseart and Nieber, 1977; Brutseart and Lopez, 1998; Zhang

et al., 2009).

The analytical solutions of Brutseart and Nieber (1977) for hydrogeological parameter

estimations were derived based on assumptions of specific initial and boundary conditions

of an ideal aquifer (Szilagyi and Parlange, 1998) in relatively humid and slowly draining

catchments settings with moderate topography. Many efforts have been made to improve

the applicability of the analytical solutions to a broader range of conditions (i.e., finite

length aquifer, vertical heterogeneity aquifer, sloping aquifer, and arid setting). For

example, Parlange et al. (2001) derived a single analytical approximate solution for Boussinesq equation that unifies the early-time and late-time solutions for finite length aquifer. It was then further improved by Mendoza et al. (2003) to derive the non-linear solutions in a drier and steeper landscape, where they used it for estimating hydraulic

parameters at semi-arid mountainous watersheds in Mexico (Mendoza et al., 2003). Rupp and Selker (2005) derived an analytical solution for the vertically heterogeneous aquifer on assumption of a power-law decreasing of $K$ from aquifer surface to bottom. Gao et al. (2013) found that considering the high variability of vertical $K$ could increase the reliability of the estimated values of $f$ and vertical mean $K$ at catchment scale. As an important

component of hydraulic gradient, the slope of the aquifer is also crucial to accurately express the groundwater flow behavior. Extended analytical solutions from horizontal to sloping aquifer were derived by Brutsaert (1994) and Hogarth et al. (2014) in different ways. Pauritsch et al. (2015) claimed that analytical models for sloping aquifers can narrow the ranges of the estimated values of $K$ and $D$ compared to that for horizontal aquifers in

complex and heterogeneous aquifer systems in an alpine catchment.

Streamflow recession analysis for determining hydrogeological parameters is popularly based on two-stages flow recessions (Brutsaert, 2005; Troch et al., 2013) in two distinct flow regimes: the early-time and the late-time regimes. Correspondingly, two equations are derived to describe the flow recessions. Their solutions could be only used to estimate two

of the three unknown parameters ($K$, $D$, and $f$). It means that one parameter must be given a priori. For example, Zhang et al. (2009) estimated catchment-scale $K$ and $f$ with given $D$. Vannier et al. (2014) estimated $K$ and $D$ with given $f$. In most of the catchments, all of the parameters are unknown. If the parameters (usually $D$ or $f$) are set in advance with certain



arbitrary values, it would bring great uncertainty of the estimated parameters.

Among the three parameters, $K$ and $f$ are dependent on soil properties. The drainable

porosity ($f$), defined as volume of water that a saturated soil will yield by gravity to the

total volume of the rock or soil (Bear, 1972), is directly dependent on soil pore connected,

soil particle-size distribution, and the soil pore structures (Ding et al., 2016). Saturated

hydraulic conductivity ($K$) can be expressed as a function of the hydraulic radius of soil

pore (defined as soil pore volume over the pore-solid surface area), indicating that $K$

depends on soil particle size, soil porosity, and pore shape factor and tortuosity (Zhang and

Schaap, 2019). Evidently, both $K$ and $f$ can be estimated from easy-to-measure soil

properties, like soil texture, bulk density, and organic content (Saxton and Rawls 2006;

Stephens et al., 1998; Zhang et al., 2016), popularly using the pedotransfer functions (PTFs)

(Saxton and Rawls 2006).

As both $K$ and $f$ depend on soil physical properties, $K$ and $f$ are significantly correlated.

Such hypothesis has been validated from field experiments (Nyberg, 1995 and Espeby,

1990) and physically based equations, including the Kozeny–Carman equation (Carman,

1937) and its modified equations (Ahuja et al., 1984, 1989; Rawls et al. 1993), which relate

$K$ to effective porosity (equal to saturated water content minus field capacity) that is viewed

identical to $f$ for shallow homogenous unconfined aquifers (Bear, 1972). Therefore, if the

analytical solutions of Boussinesq equation combine the modified Kozeny–Carman

equation, the three unknown parameters ($K$, $D$, and $f$) can be determined simultaneously by

using streamflow recession analysis.

The aim of this study is to extend the estimations of catchment-scale hydrogeological

parameters ($K$, $f$, and $D$) from two to three parameters by using assessable information of





catchment characteristics and observed streamflow. It is achieved by an approach deriving
the analytical expressions of the parameters $K, f,$ and $D$ in terms of the analytical solutions
of Boussinesq equations combining with the modified Kozeny–Carman equation. The

approach is then applied to four experimental catchments with detailed observations of
catchment properties and measurements of the parameters. The sensitivity analysis is
executed to reveal the main controlling factors in the determination of the three
hydrogeological parameters.

## 2 Methodology

### 2.1 Analytical solutions of Boussinesq equation for sloping aquifer

In hilly areas, a catchment can be viewed as an assembly of a series of sloping hillslopes
along with river networks (Fig. 1a). To analytically estimate catchment-scale
hydrogeological parameters, the non-uniformly distributed sloping aquifers can be
simplified as an equivalently homogeneous aquifer (Fig. 1b).

The one-dimensional subsurface flow on the sloping aquifer (Fig. 1b) can be described
by

$$q_x = -K\eta(\cos(\alpha)\frac{\partial \eta}{\partial x} + \sin(\alpha)) \qquad (1)$$

where $q_x$ is the flow rate per unit width of the aquifer, $\eta$ is water table, $K$ is the saturated
hydraulic conductivity, $\alpha$ is the hillslope gradient, and $x$ is the distance from the river to

hillslope ridge.

Assuming that aquifer is isotropic and homogeneous, substituting Eq. (1) into the
continuity equation yields:

$$f\frac{\partial \eta}{\partial t} = K\left[\cos(\alpha)\frac{\partial}{\partial x}\left(\eta\frac{\partial \eta}{\partial x}\right) + \sin(\alpha)\frac{\partial \eta}{\partial x}\right] \qquad (2)$$

where $t$ is time and $f$ is the drainable porosity.



Brutsaert (1994) derived the analytical solution of flow rate in the hillslope with specific

conditions, including: initially fully saturation of the aquifer, a sudden drawdown of the

water table (hydraulic head $\eta$) to the channel as outflow starts for the flow condition at $x=0$

(Fig. 1b), and the infinitely wide aquifer that ensures negligibility of draining flow

influence on the upper boundary of the aquifer. With that, the analytical solution of Eq. (2)

for unit width hillslope is

$$q_{xf} = [(p/\pi \cos(\alpha)KfD^3)]^{1/2}t^{-1/2} \tag{3}$$

where $q_{xf}$ is fast flow in the early-time recession, $p$ is a constant equal to 0.3465, and $D$ is

the aquifer thickness.

When the water table at the boundary ($x=B$, where $B$ is the hillslope length) starts falling,

the analytical solution of slow flow ($q_{xs}$) in the late-time recession is

$$q_{xs} = \frac{2pcos(\alpha)KD^2}{B}\frac{z_1{}^2[1-2cos(z_1)\exp(Hi/2)]}{(z_1{}^2+Hi^2/4+\frac{Hi}{2})}exp\left[-\frac{(z_1{}^2+Hi^2/4)pcos(\alpha)KD}{fB^2}t\right] \tag{4}$$

where $Hi = B \tan \alpha/(pD)$; $z_1 = \pi/2$ for small $Hi$ in gentle hillslopes and thick aquifers,

while $z_1 = \pi$ for large $Hi$ in steep hillslopes and thin aquifers.

**2.2 Recession analysis**

In the flow recession period, the relationship between streamflow ($Q$) and change of

streamflow ($-dQ/dt$) was proposed by Brutsaert and Nieber (1997)

$$-\frac{dQ}{dt} = f(Q) \tag{5}$$

The function $f(Q)$ is often expressed by power-law equation as:

$$-\frac{dQ}{dt} = aQ^b \tag{6}$$

where $a$ and $b$ are constants for a specific catchment. As streamflow in catchment outlet

can be viewed as the collection of flow from hillslopes along the river, $Q$ is the multiple of





two-fold $q_x$ and river length ($L$). Inserting Eq. (3) into Eq. (6), the constants $a$ ($a_f$) and $b$ ($b_f$)

for the early-time recession are obtained as:

$$a_f = (8p/\pi \cos \alpha)^{-1}(KfD^3L^2)^{-1} \tag{7}$$


$$b_f = 3 \tag{8}$$

Inserting Eq. (4) into Eq. (6), the constants $a$ ($a_s$) and $b$ ($b_s$) for the late-time recession are

$$a_s = \pi^2 p \cos \alpha (4 + \left(\frac{Hi}{\pi}\right)^2)KD(L/A)^2/f \tag{9}$$

$$b_s = 1 \tag{10}$$

When the term $\left(\frac{Hi}{\pi}\right)^2$ is much larger than 4, the subordinate term of 4 in the bracket in the

right side of Eq. (9) can be ignored. Thus, Eq. (9) can be expressed as

$$a_s = \pi^2 p \cos \alpha \left(\frac{Hi}{\pi}\right)^2 KD(L/A)^2/f \tag{11}$$

For capturing recession processes from the streamflow hydrographs, the selected flow

events should recess markedly at least 4 days after rainfall ceases. Recession data in the

first day are excluded for eliminating the influence of direct and surface flow. Because the

observed recession rate $-\frac{dQ}{dt}$ is one to two orders less than streamflow $Q$, discretization

errors on $-\frac{dQ}{dt} \sim Q$ plot are vulnerable to the measurement noise, especially in log-log space

(Gao et al., 2017). The time-derivative of streamflow $\frac{dQ}{dt}$ and the concurrent streamflow $Q$

between the time interval ($i$, $i$-$j$) are calculated in terms of the method proposed by Rupp

and Selker (2006)


$$-\frac{dQ}{dt} \approx \frac{Q(i)-Q(i-j)}{t(i)-t(i-j)}, \quad i = 2, 3, \dots, N; \ 1 \le j \le i - 1 \tag{12}$$

$$Q \approx \frac{1}{(j+1)}\sum_{k=i-j}^{i} Q(k) \tag{13}$$

where $i$ represents data points taken at discrete time increments and $j$ is the number of time


increments over which $-\frac{dQ}{dt}$ is calculated. Thus, a variable time interval $t(i) - t(i - j)$ is used to properly scale the observed drop in streamflow in order to avoid artifacts in data.

Additionally, if values of recession segments in contiguous streamflow data are equal, only the latest data are involved in the calculations using Eqs. (12) and (13).

**2.3 Modified Kozeny–Carman equation relating $K$ to $f$**

Kozeny (1927) derived a power-law function that relates the saturated hydraulic conductivity $K$ to soil porosity based on the Hagen-Poiseuille's equation. It was later

modified as the Kozeny–Carman equation after Carman (1937, 1956), which accounts for tortuosity in tube flow. Ahuja et al. (1984) proposed a modified Kozeny-Carman equation relating $K$ to effective porosity, which is identical to drainable porosity $f$ in this study. The modified Kozeny-Carman equation is in the form of $K = Cf^m$, where $C$ and $m$ are constants. To obtain estimated values for coefficients $C$ and $m$, one needs to fit the equation

with measured data of $K$ and $f$ at a variety of catchment sites. Rawls et al. (1998) redefined the exponent $m$ as 3 minus the Brooks-Corey pore size distribution index ($\lambda$):

$$K = \gamma \times f^{3-\lambda} \tag{14}$$

where $\gamma$ is equal to $5.36 \times 10^{-4}$ for $K$ in a unit of m/s, obtained by fitting the measured data for 26 soil texture/porosity classes. The value $\lambda$ ranges $0.165 \sim 0.694$, estimated by

fitting a log-log curve between water content and pressure head using the $-33$ and $-1500$ kPa water contents (Rawls et al., 1993), according to USDA soil texture classes and over 900 measurements. Eq. (14) has been fitted by a variety of soils, making the equation universally applicable.

**2.4 Estimation of catchment-scale hydrogeological parameters**

Substituting Eq. (14) into Eq. (7) gives



$$a_f^{-1} = C_f K^{1+\beta} D^3 \qquad (15)$$

where $C_f = 8p/\pi\cos\alpha L^2 \gamma^{-\beta}$ , and $\beta = 1/(3-\lambda)$. Similarly, substituting Eq. (14) to Eq. (9) gives

$$a_s = C_{s1}(4 + C_{s2}D^{-2})K^{1-\beta}D \qquad (16)$$

where $C_{s1} = B^{-2}\gamma^{\beta}\pi^2 p\cos\alpha/4$ , and $C_{s2} = B^2\tan^2\alpha/(\pi^2 p^2)$. Combining Eq. (15) with Eq. (16), $K$ and $D$ can be obtained from implicit equations as follows

$$4D^{\frac{-2+4\beta}{1+\beta}} + C_{s2}D^{\frac{-4+2\beta}{1+\beta}} = (a_s C_{s1}^{-1})(a_f C_f)^{\frac{1-\beta}{1+\beta}} \qquad (17)$$

$$4K^{\frac{2-4\beta}{3}} + (a_f^{\frac{2}{3}} C_f^{\frac{2}{3}})C_{s2}K^{\frac{4-2\beta}{3}} = (a_s C_{s1}^{-1})(a_f C_f)^{\frac{1}{3}} \qquad (18)$$

According to Eq. (14), drainable porosity (*f*) can be expressed as

$$f = r^{-\beta}K^{\beta} \qquad (19)$$

Hence, *f* can be obtained easily when $K$ is estimated from Eq. (18). This proves that the essential catchment-scale hydrogeological parameters ($K$, *f*, and $D$) can be estimated simultaneously by combining the analytical solutions from Boussinesq equation with the modified Kozeny–Carman equation (Eqs. (17), (18), and (19)).

**3 Study catchments and data**

The proposed approach for catchment-scale hydrogeological parameter estimation was tested in four experimental catchments in the northern hemisphere. These include, the relict Schöneben Rock Glacier (SPG) in Austria, the Hemuqiao experimental catchment (HMQ) in China, the Panola Mountain Research Watershed (PMRW) and the WS10 from HJ

Andrews Experimental Forest in the USA (Fig. 2). All of the four catchments are located in humid climate region with mean annual precipitation greater than 1000 mm. The experimental catchments are small and steep, with areas ranging from 0.102 to 1.35 km²



and average catchment hillslope gradients varying from 10.2° to 29°. The uniformly

distributed soil deposits in all four catchments falls under typical soil types (Table 1).

The daily observation data of streamflow and precipitation at PMRW, HMQ, and WS10

are available. The observation data at PMRW are available during 1986-2016,

(https://doi.org/10.5066/P94JC2PD). As a part of HJ Andrews catchment, WS10 has

observation data during 2000 ~ 2011 (https://andrewsforest.oregonstate.edu/). Located in

one of the famous hydrologic experimental catchments (Jiangwan) in China before the

1970s, HMQ has observation data during 1957 ~1958 published in China hydrologic year

book. As to SPG, data published in a study by Pauritsch et al. (2015) are directly used in

our analysis.

All the four catchments have field measures of $K$ and $D$, which are used for validating

the estimates of our proposed methods. In PMRW, soil $K$ values were measured at two soil

pits using constant head permeameters method by McIntosh et al. (1999), while $K$ values

for saprolite and bedrock were measured by falling head permeameter (White et al., 2002).

In HMQ, the $K$ values were measured at three typical soil pits by using falling head

permeameter (Han et al., 2016). In WS10, vertical $K$ values were measured at ten soil pits

by using constant head permeameter (Harr, 1977). For SPG, the catchment equivalent

values of $K$, $f$, and $D$ were obtained from Pauritsch et al. (2015) according to field

measurements (Winkler et al., 2016).

Although the aquifer deposits in each of these catchments are dominated by one type of

soil textures (Table 1), the point scale hydrogeological parameters observations show great

heterogeneity, especially for saturated hydraulic conductivity $K$ and soil/saprolite thickness

$D$ (Table 2). $K$ shows highly vertical variations. The value of $K$ at the upper soil layer can



be one order larger than the bottom one or the underlain saprolite (Table 2). Across

catchments, the soil thickness $D$ differs but the riparian areas much thicker than that the

hillslopes for all catchments. For example, the soil thickness $D$ for the typical soil profiles

in PMRW ranges 0.6 ~ 1.6 m on hillslope but reaches 5 m at the riparian area (Peters et al.,

2003). The average soil thickness on a typical hillslope in HMQ is 0.8 m (Han et al., 2018)

while the soil in riparian with colluvium and residuum deposits can reach up to 6 m (Han,

et al., 2016). Saprolite or regolith are an important part of groundwater storage (Hale et al.,

2016), they can be much thicker than the upper soils and more spatially heterogeneous. In

WS10, the thickness of saprolite is 3.7 m in average, varying from 1 to 7 m (McGuire et

al., 2010), nearly three times greater than the average thickness of soils (1.30 m). In PMRW,

spatial variability of the saprolite thickness (0 ~ 5 m) over granodiorite (Peters et al., 2003)

is much greater than that of the upper soil thickness (0.6 ~ 1.6 m).

Additionally, values of $K$ and $f$ for each catchment are estimated in terms of soil texture

by using pedotransfer function proposed by Saxton and Rawls (2006) (Table 2).

**4 Results**

**4.1 Estimated recession constants**

Plots of recession data in the form of $-\frac{dQ}{dt} \sim Q$ for each of the four catchments are shown

in Fig. 3. In PMRW, HMQ, and WS10, the lower envelopes with 10% of data points are

excluded in order to remove the effect of outliers on the envelope lines (Brutsaert and

Lopez, 1977). For SPG, the recession data are adopted directly from Pauritsch et al. (2015).

Here, only winter data are used to fit the lower envelop lines as suggested by Pauritsch et

al. (2015). The lower envelop lines with slopes of 3 and 1are used to derive the recession

intercepts for early- and late-time recessions ($a_f$ and $a_s$), respectively.





The estimated constants of $a_f$ and $a_s$ in all catchments are listed in Table 3. The values

of $a_f$ range in $3.0\times10^{-4}\sim 9.11\times10^{-2}$ m$^{-6}$ s, where the largest is estimated for WS10 and

smallest is estimated SPG. The value of $a_f$ in WS10 is about one order of magnitude larger

than the value for SPG and PMRW, and twice larger than the values for HMQ. The values

of $a_s$ range in $2.36\times10^{-7}\sim 1.43\times10^{-6}$ s$^{-1}$ (8.1 $\sim$ 49.1 days), the largest in HMQ and smallest

in SPG. The timescales of late-time recessions $(1/a_s)$ in SPG, PMRW, HMQ, and WS10

are 49.0, 35.6, 8.1, and 23.1 days, respectively. Brutseart (2008) claimed that the timescale

of the late-time recession commonly ranges $45 \pm 15$ days for mesoscale catchments.

Compared to this range, the late-time recessions are relatively fast in our catchments except

for SPG where the soil deposits are much thicker.

**4.2 Estimated catchment-scale hydrogeological parameters**

Given catchment physical properties (area, river length, and slope in Table 1), the estimates

of $a_f$ and $a_s$ as well as the soil texture-based $\lambda$ (Table 3), the catchment-scale

hydrogeological parameters ($K$, $D$, and $f$) are calculated according to solutions from

combinations of Eqs. (17), (18), and (19). The estimated hydrogeological parameters vary

significantly across catchments, especially for $K$ and $D$ in Table 3. The estimated values of

$K$ range from $1.75\times10^{-6}$ to $2.36\times10^{-5}$ m/s. The smallest $K$ value was observed in WS10, it

is about one order of magnitude smaller than that of PMRW, SPG, and HMQ. The

estimated $f$ ranges in $0.13 \sim 0.28$, and aquifer thickness $D$ ranges in $2.99 \sim 18.7$ m for the

four catchments. Three of the catchments have the thickness $D$ in several meters except for

SPG where the aquifer is much thicker ($D = 18.7$ m). The differences of either $K$ or $f$ values

in the four catchments are highly attributed to the soil texture. As expected, clay loam in

WS10 has the smallest $K$ and $f$ while equivalent loamy sand in SPG has the largest $K$ and $f$



(Table 1).

To validate the reliability of our catchment-scale hydrogeological parameter estimation approach, the estimated values of $K$, $f$, and $D$ are compared to the measured values in all

four catchments. Additionally, the estimated values of $K$ and $f$ are compared to the estimates from soil texture (Saxton and Rawls 2006) (Table 2 and Fig. 4). For SPG, the estimated catchment-scale $K$ ($2.36{\times}10^{-5}$ m/s) is close to the catchment equivalent value ($1.00{\times}10^{-5}$ m/s) given by Pauritsch et al. (2015) in terms of the order of magnitude. The estimated value of $K$ for PMRW ($1.90{\times}10^{-5}$ m/s) is within the range of measured values

between upper soil ($1.79{\times}10^{-4}$ m/s) and underlying saprolite ($5.0{\times}10^{-6}$ m/s). In HMQ, the estimated value of $K$ ($1.20{\times}10^{-5}$ m/s) is within the range of measured values in the whole soil profile ($2.5{\times}10^{-6} \sim 5.09{\times}10^{-5}$ m/s) but closer to the measures within the profile depths of $30 \sim 90$ cm. The estimated catchment-scale $K$ is 75% larger than the geometric mean of the measured $K$ in the vertical profile for these three catchments. An exception is that the

estimated $K$ value ($1.75{\times}10^{-6}$ m/s) in WS10 is much below the lower limit of the measured values in a shallow soil profile ($4.4{\times}10^{-5} \sim 1.1{\times}10^{-3}$ m/s). The two orders of magnitude smaller $K$ value estimated in WS10 could be attributed to the fact that the measurements were only taken from the upper soils (1.5 m in maximum) with abundant macropores (Harr, 1977), which could be much greater than that in the soil below (e.g. $\sim$5.77 m thickness in

our estimation). Fig. 4b shows the estimated catchment-scale $K$ in this study vs. the estimated $K$ from soil texture. Both estimated $K$ values are close to the 1:1 line, indicating that both estimates are in agreement. The $f$ values from our study method are close to the estimated $f$ from soil texture with slight overestimation by $4\% \sim 47\%$ (Fig. 4c, an average of 23.7%).





The estimated aquifer thickness $D$ values in our study are comparable to the measured

values in SPG (Fig. 4d). But PMRW and HMQ showed significantly larger than soil

thickness on the hillslope, the magnitudes are within the range of mean soil thickness on

hillslopes and the maximum soil thickness in riparian. In WS10, the estimated $D$ is close

to the measured $D$ when both soil and saprolite are considered as effective groundwater

aquifers.

**4.3 Sensitivity analysis for the hydrogeological parameters**

As $\left(\frac{Hi}{\pi}\right)^2$ in Eq. (9) is proven to be much larger than 4 (Table 3), Eq. (10) is adopted by

neglecting the subordinate term on the right side of Eq. (9). Thus, the hydrogeological

parameters ($K$, $D$, and $f$) in Eqs. (17) ~ (19) can be explicitly expressed as

$$D = \left(C_D a_f^{-(1-\beta)} a_s^{-(1+\beta)} (\tan\alpha)^2 (\sin\alpha)^{2\beta} \gamma^{2\beta} L^{-2(1-\beta)}\right)^{\frac{1}{4-2\beta}} \tag{20}$$

$$K = \left(C_K a_f^{-1} a_s^{3} (\tan\alpha)^{-2} (\sin\alpha)^{-4} \gamma^{-2\beta} L^{-2}\right)^{\frac{1}{4-2\beta}} \tag{21}$$

$$f = \left(C_K a_f^{-1} a_s^{3} (\tan\alpha)^{-2} (\sin\alpha)^{-4} \gamma^{-4} L^{-2}\right)^{\frac{\beta}{4-2\beta}} \tag{22}$$

where $C_D = (8p/\pi)^{-1+\beta}(4p)^{-(1+\beta)}$, $C_K = 8\pi p^2$, and $\beta = \frac{1}{3-\lambda}$ ranging in 1/3 and 1/2.

Eqs. (20) ~ (22) show that the hydrogeological parameters are controlled by not only the

variables of streamflow recessions ($a_s$ and $a_f$) but also catchment topography and

topological properties (slope $\alpha$ and river length $L$) as well as parameters of the modified

Kozeny–Carman equation ($\gamma$ and $\lambda$). Given a 10% change of each influence variable relative

to the estimated or known values in Tables 1 ~ 3, the sensitivity of $D$, $K$, and $f$ to the

influencing variables can be estimated. Here, as an example, the sensitivity analysis is done

for HMQ catchment (Fig. 5). It clearly shows that $K$ and $f$ decrease with increases of the

independent parameters except for $a_s$. $D$ decreases with $a_f$, $a_s$, and $L$, but it increases with





$\alpha, \lambda$, and $\gamma$. Among all the independent variables, the catchment slope $\alpha$ is the most sensitive variable to the three hydrogeological parameters (Fig. 5). The next sensitive variable is late-time recession rate ($a_s$) for $K$ and $D$, and $\gamma$ for $f$. The most insensitive variables are $\lambda$

and $a_f$ for $f$ and $D$, and, $\lambda$ and $\gamma$ for $K$. Among the three parameters, $K$ is the most sensitive parameter to the independent variables ($\alpha, a_s, a_f$, and $L$).

Overall, accurate representation of catchment aquifer slope $\alpha$ and analysis of late-time recession could increase the reliability of the three estimated parameters, especially for $K$.

## 5 Discussions

**5.1 Comparison of parameter estimates from analytical solutions in horizontal and sloping aquifers**

Based on our sensitivity analysis, in the presented simultaneous estimation approach of the three hydrogeological parameters, catchment slope ($\alpha$) exerts considerable influence on the parameters. To evaluate effects of the catchment slope on recession analysis, we compared

the estimated values of the results from horizontal aquifer assumption. For a horizontal aquifer, where $K$ and $f$ can be estimated from equations as follows (Brutseart, 2005):

$$K = 0.5757(L^2/A)^{-1}D^{-2}(a_s/a_f)^{1/2} \qquad (23)$$

$$f = 1.9688(DA)^{-1}(a_s a_f)^{-1/2} \qquad (24)$$

Combing Eqs. (7) with (9), $K$ and $f$ for sloping aquifer can be expressed as

$$K = 0.5757(L^2/A)^{-1}D^{-2}\alpha(4 + (\tfrac{Hi}{\pi})^2)^{-1/2}(a_s/a_f)^{1/2} \qquad (25)$$

$$f = 1.9688(DA)^{-1}(4 + (\tfrac{Hi}{\pi})^2)^{1/2}(a_s a_f)^{-1/2} \qquad (26)$$

Thus, $K$ and $f$ varying with $D$ can be described by Eqs. (23) and (24) for horizontal aquifers and by Eqs. (25) and (26) for sloping aquifers. Here, $K$ and $f$ related to $D$ are given in two of the catchments, HMQ and PMRW, where catchment slopes are taken as representatives





from the mild (10.2° in PMRW) to the steep (20° in HMQ). As shown in Fig. 6, *K* and *f*

        decrease with *D* for both sloping and horizontal aquifers. The differences of *f* and *K*

        between sloping and horizontal aquifers are tremendous. The estimated values of *f* from

        the horizontal aquifer are about one order of magnitude smaller than those from sloping

        aquifer while the estimated values of *K* from the horizontal aquifer are about one order of

magnitude greater than those from a sloping aquifer. Estimates of catchment-scale *f* from

        horizontal aquifer assumption are too small to be reasonable. For example, *f* is 0.0083 for

        *D*=2.99 m in HMQ, and 0.019 for *D*=4.84 in PMRW. Meanwhile, *K* estimated from

        horizontal aquifer assumption reaches a larger magnitude of over $10^{-4}$ m/s when *D* is less

        than 5.5 m in HMQ and less than 8 m in PMRW. This order of magnitude of *K* value

approaches the macropore flow velocity (in the order of $10^{-4} \sim 10^{-3}$ m/s) at hillslope scale

        according to the summary from 110 observations by Gao et al (2018). Whilst the measured

        *K* values are in the order of magnitude of $10^{-5}$ and $10^{-6}$ m/s in these four catchments. Thus,

        *K* could be extremely overestimated by horizontal aquifer assumption.

**5.2 Effect of vertical heterogeneity of soils on the determination of *K* and *f***

For natural soils, *K* usually decreases with soil depth (Brooks and Boll, 2004) and field

        measurements also indicate that *f* could decrease with soil depth as well (Harr, 1977;

        McGuire et al., 2007). The decrease of *K* with depth could reduce the late-time recession

        rate as the water level becomes lower. The depth dependent *K* affects estimates of

        hydrogeological parameters from streamflow recessions. According to our previous study

(Gao et al., 2013), the estimated average *K* in a vertical profile increases with vertical

        variability of *K* in terms of the analytical solutions of Boussinesq equation considering the

        vertical decrease of *K* (Rupp and Selker, 2005). The measured *K* in our selected catchments





(e.g. PMRW, HMQ, and WS10) shows that $K$ at the upper soil can be $1 \sim 2$ orders of magnitude greater than that in the deep deposits. The estimated $K$ from our proposed

method can be regarded as an equivalent value that represents $K$ in a specific range of the soil profile. For example, in PMRW, the estimated value of $K$ is more likely to represent the equivalent value between soil and saprolite. In HMQ, the estimated value of $K$ is closer to the measures within the soil profile at $30 \sim 90$ cm depths even though the soil profiles on the hillslope can be as deep as $1.2 \sim 4.4$ m (Fig. 4a). The ranges of the measured $K$ cover

our estimated values in three catchments, indicating that the measured ranges of $K$ can be used to restrict parameter variations in model calibration. However, it should be mentioned that the selected profiles and sites for measures should be representative of a catchment. This study shows that the field measurements only taken at the upper soils (1.5 m in the maximum depth) cannot capture the whole profile values of 5.77 m in depth in WS10.

**5.3 Effect of the riparian zone on flow recessions**

Burns et al. (2001) demonstrated that the hillslopes contribute flow mainly during large storm event while the riparian zone in the gentle areas with thick deposits of sediment takes a vital role on runoff generation and flow regulation (Clark et al., 2009; McGlynn and McDonnell, 2003; Klaus et al., 2015). A riparian zone can generate a vast majority of runoff

during small event and between events (recession processes) (Burns et al., 2001; McGlynn and McDonnell, 2003), and contribute a primary flow component throughout the recessions (Clark et al., 2009). Our estimated catchment-scale aquifer thickness ranges between the measured thicknesses in hillslope and riparian zone but is much thicker than the thickness in hillslope areas. It implies that thicker deposits in the riparian zone have involved to slow

down flow recessions since the thicker aquifer reduces hydraulic conductivity $K$ as shown


in Fig. 6. Thus, detail investigations on the riparian zone properties, such as the deposit thickness and hydrologic connectivity with hillslopes and streams are vital for appropriate analysis of flow recession behaviors.

## 6 Conclusions

Catchment-scale hydrogeological parameters ($K$, $f$, and $D$) are important for acquiring catchment hydrologic properties and simulating hydrologic processes, especially low flow processes. In our study, a new method that combines analytical solutions of Boussinesq equation with the modified Kozeny–Carman equation is proposed to estimate three key hydrogeological parameters ($K$, $f$, and $D$) simultaneously at catchment scale. Four

experimental catchments are used to test the validity of the proposed method.

    The new method only needs streamflow observations and catchment geometric characteristics to estimate catchment-scale parameters. Compared to the field measurements and the estimated values from soil texture, our estimated results range within the measured values at various sites/profiles in the catchments and can be regarded as the

equivalent values in specific depths of the aquifer. It indicates that our estimations can be used as model parameters for simulation of the catchment-scale hydrologic processes, particularly baseflow processes.

    Details of catchment geometric characteristics can affect the parameter estimation accuracy. The sensitivity analysis of the parameters to the catchment characteristics and

recession constants showed that catchment slope is the most sensitive variable to the three hydrogeological parameters. Neglecting effects of the catchment slope on streamflow recession analysis would result in significant overestimation in saturated hydraulic conductivity and underestimation in drainable porosity. Additionally, the estimated aquifer

thickness is much greater than the measured thickness on hillslopes but close to the

thickness in the riparian zone, indicating that riparian zone could take an important role on

flow recessions even in the hilly catchments.

*Data availability.* The observation data of streamflow and precipitation at PMRW and

WS10 catchments are available at https://doi.org/10.5066/P94JC2PD and

https://andrewsforest.oregonstate.edu/, respectively. The original observation data of

streamflow and precipitation in SPG and HMQ catchments cannot be publicly accessed.

*Author contributions.* MG and XC designed research; MG performed research; JL provided

data and involved in derivation of the theoretical equations. MG wrote the paper. XC and

JL extensively edited the paper.

*Competing interests.* The authors declare that they have no conflict of interest.

**Acknowledgments**

This research was supported by the National Natural Scientific Foundation of China (NSFC)

(grant numbers 41901029, 91747203, and 91647108) and Independent Innovative

Foundation of Tianjin University (grant number 2019XZS-0016). We thank Mr. Brent T

Aulenbach for providing the discharge and precipitation data of PMRW and Guta Abeshu

for editing the manuscript.

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





# Tables

**Table 1.** The landscape and climate properties of the four experimental catchments

| | Catchment | Location | Area (km²) | Slope (°) | River length (km) | Elevation (m) | Precipitation (mm/a) | Soil | Geology |
|---|---|---|---|---|---|---|---|---|---|
| 1 | Schoneben rock glacier (SPG) | Styria, Austria | 0.67 | 15 | 0.30[*] | 1715 ~ 2295 | 1190 | Loamy Sand[**] | Coarse-grained gneissic sediments |
| 2 | Panola Mountain Research Watershed (PMRW) | Georgia, USA | 0.41 | 10.2 | 0.48[**] | 222 ~ 279 | 1225 | Sandy loam | Granodiorite (predominantly) |
| 3 | Hemuqiao Experimental Catchment (HMQ) | Zhejiang, China | 1.35 | 20 | 2.50[**] | 160 ~ 600 | 1580 | Silt loam | Quartz sandstone |
| 4 | HJ Andrews Experimental Forest (WS10) | Oregon, USA | 0.10 | 29 | 0.58[**] | 473 ~ 680 | 2350 | Clay loam | Volcaniclastics (predominantly) |

Comments: [*] Equivalent value from Pauritsch et al. (2015).

[**] Measured from map in literature of Clark et al. (2009) for PMRW, Han et al. (2018) for HMQ, and Klaus et al. (2015) for WS10.

[**] The SPG surface sediment characteristics is relative coarse and is even described as "*covered by coarse-grained, blocky material consisting of gneissic rocks ranging from cubic decimeters to a few cubic meters*" (Pauritsch et al., 2015). Its soil texture is set as loamy sand combining this description with comparison of the hydraulic properties ($K$ and $f$) measured with related soil.



**Table 2.** Values of the hydrogeological parameters from field measurements and estimates from soil texture in the four catchments

| | Catchments | $K$ (m/s) | | $D$ (m) | $f$ (-) (Estimated from soil texture) |
| | | Measured in field | Estimated from soil texture | | |
|---|---|---|---|---|---|
| 1 | SPG | $1.00\times10^{-5*}$ | $3.00\times10^{-5}$ | $30^{*}$ | $0.2^{*}$ |
| 2 | PMRW | $1.79\times10^{-4}$ (soil); $5.0\times10^{-6}$ (saprolite); $1.6\times10^{-6}$ (bedrock) | $1.39\times10^{-5}$ | 1.1 (hillslope) ~ 5 (riparian area) | 0.27 |
| 3 | HMQ | $5.09\times10^{-5}$ (soil surface) $5.4\times10^{-6}$ (30 ~ 90 cm) $2.5\times10^{-6}$ (120 ~ 440 cm) | $4.47\times10^{-6}$ | 0.8 (hillslope) ~ 6 (riparian area) | 0.17 |
| 4 | WS10 | $3\times10^{-4}$ (soil mean) (range: $4.4\times10^{-5}$ ~ $1.1\times10^{-3}$) | $1.19\times10^{-6}$ | 5 (hillslope and saprolite) | 0.12 |

Comment: * These values are equivalent values obtained from Pauritsch et al. (2015).






**Table 3.** Estimated hydrogeological parameters at catchment scale

| | Catchments | $\lambda$ (-) | $a_f$ (m$^{-6}$s) | $a_s$ (s$^{-1}$) | Estimated hydrogeological parameters | | | $Hi$ |
|---|---|---|---|---|---|---|---|---|
| | | | | | $K$ (m/s) | $D$ (m) | $f$ (-) | |
| 1 | SPG | 0.55 | $3.0\times10^{-4}$ | $2.36\times10^{-7}$ | $2.36\times10^{-5}$ | 18.7 | 0.28 | 16.0 |
| 2 | PMRW | 0.38 | $8.50\times10^{-3}$ | $3.25\times10^{-7}$ | $1.90\times10^{-5}$ | 4.84 | 0.28 | 57.7 |
| 3 | HMQ | 0.23 | $2.40\times10^{-3}$ | $1.43\times10^{-6}$ | $1.20\times10^{-5}$ | 2.99 | 0.25 | 95.2 |
| 4 | WS10 | 0.24 | $9.11\times10^{-2}$ | $5.00\times10^{-7}$ | $1.75\times10^{-6}$ | 5.77 | 0.13 | 24.4 |



## Figure captions

**Figure 1.** Schematic representation of (a) catchment and (b) the cross section of an

unconfined aquifer lying on a sloping impermeable layer.

**Figure 2.** Locations of the four experimental catchments adopted in this study.

**Figure 3.** Recession analysis for estimation of recession parameters $a_f$ and $a_s$ in (a) SPG,

(b) PMRW, (c) HMQ, and (d) WS10 (the figure (a) is modified from Fig. 3 in Pauritsch et

al. (2015)).

**Figure 4.** The estimated catchment-scale hydrogeological parameters in this study vs. the

measured values in field and the estimated values from soil texture. Note: (a) estimated $K$

in this study vs. measured $K$, (b) estimated $K$ in this study vs. estimated $K$ from soil texture,

(c) estimated $f$ in this study vs. estimated $f$ from soil texture, and (d) estimated $D$ in this

study vs. measured $D$.

**Figure 5.** Sensitivity analysis of the estimated parameters ($K$, $D$, and $f$) in terms of the

relative changes of the parameters caused by 10% increase of any independent variable in

HMQ.

**Figure 6.** Comparison of the estimated $K$ and $f$ from sloping and horizontal aquifer with

changing $D$ in (a) PMRW and (b) HMQ.




# Figures

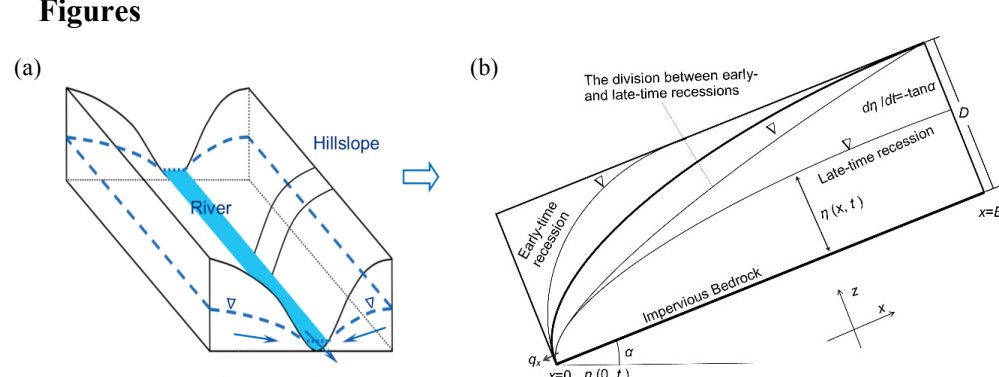

**Figure 1.** Schematic representation of (a) catchment and (b) the cross section of an

615          unconfined aquifer lying on a sloping impermeable layer.





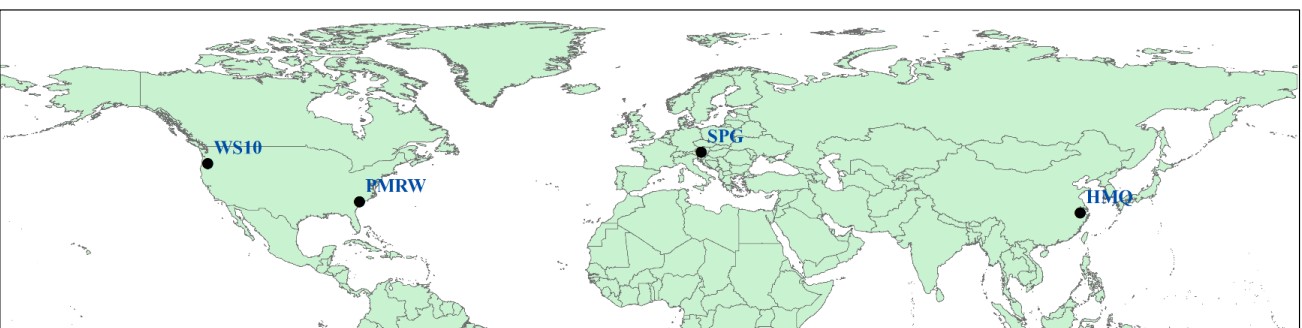

**Figure 2.** Locations of the four experimental catchments adopted in this study.





(a)


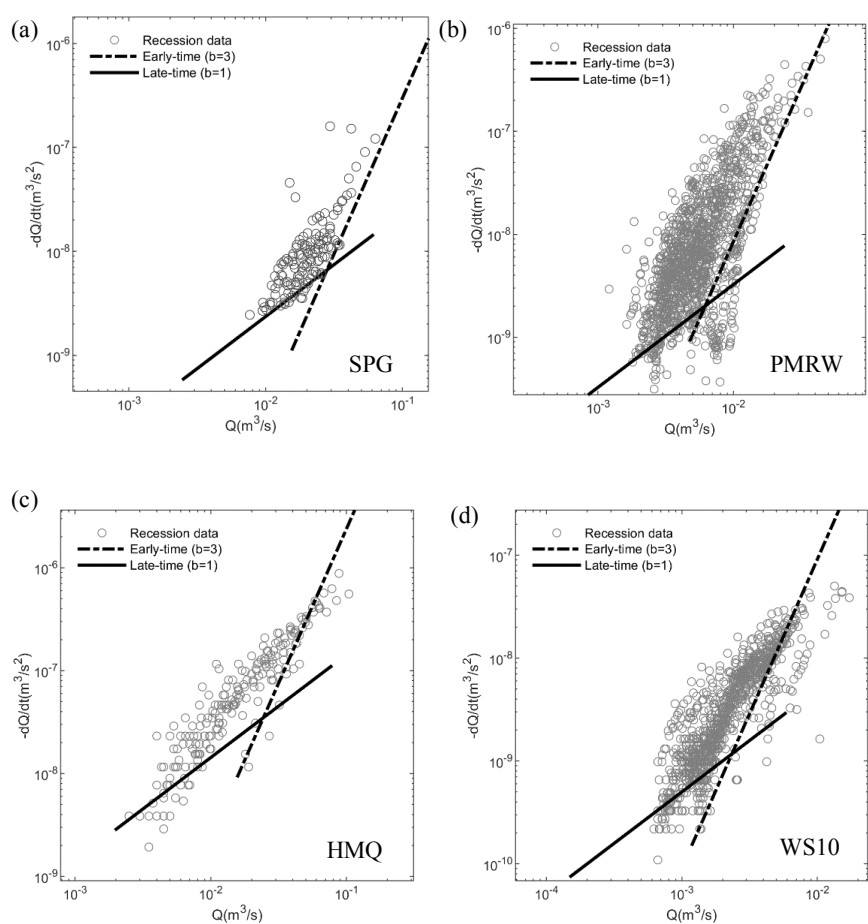

**Figure 3.** Recession analysis for estimation of recession parameters $a_f$ and $a_s$ in (a) SPG,

(b) PMRW, (c) HMQ, and (d) WS10 (the figure (a) is modified from Fig. 3 in Pauritsch et

al. (2015)).

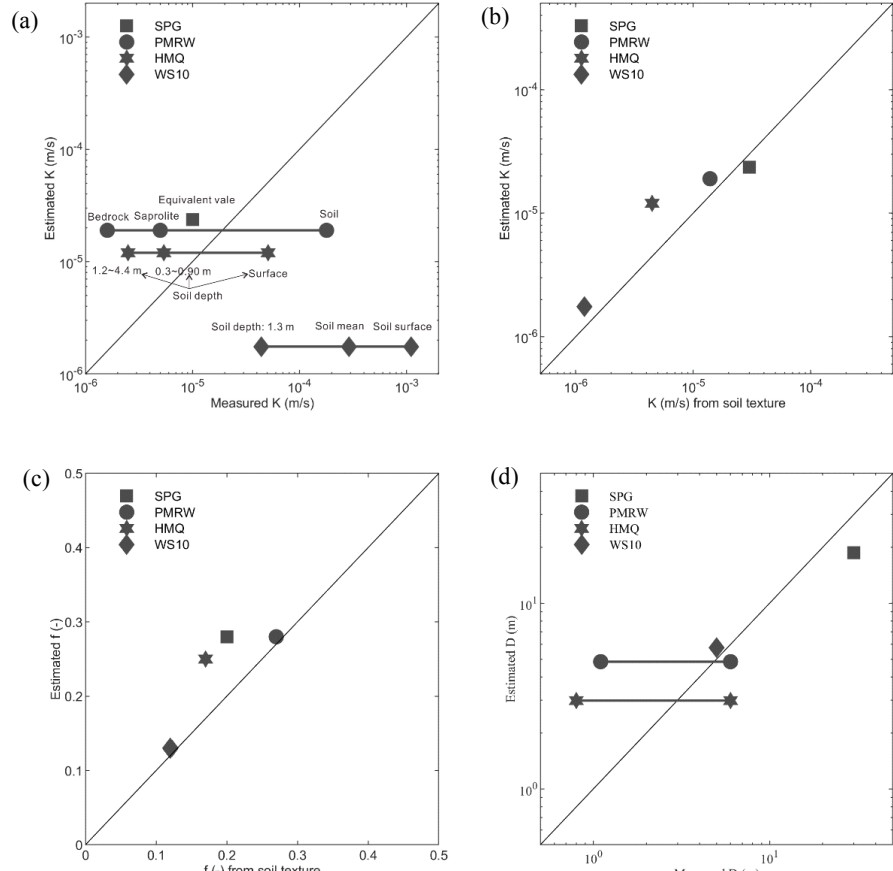


**Figure 4.** The estimated catchment-scale hydrogeological parameters in this study vs. the measured values in field and the estimated values from soil texture. Note: (a) estimated $K$ in this study vs. measured $K$, (b) estimated $K$ in this study vs. estimated $K$ from soil texture, (c) estimated $f$ in this study vs. estimated $f$ from soil texture, and (d) estimated $D$

in this study vs. measured $D$.



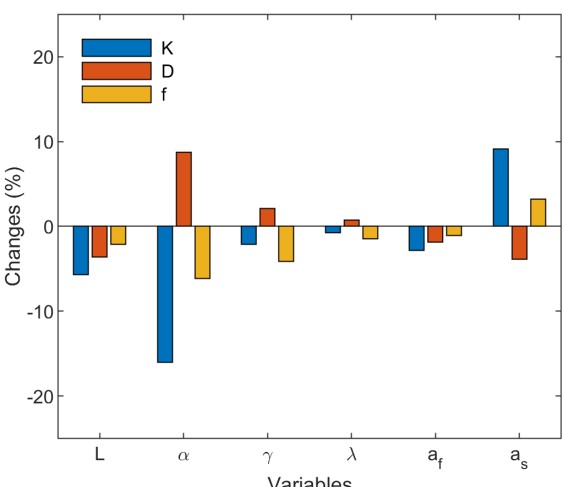

**Figure 5.** Sensitivity analysis of the estimated parameters (*K*, *D*, and *f*) in terms of the
relative changes of the parameters caused by 10% increase of any independent variable in
HMQ.





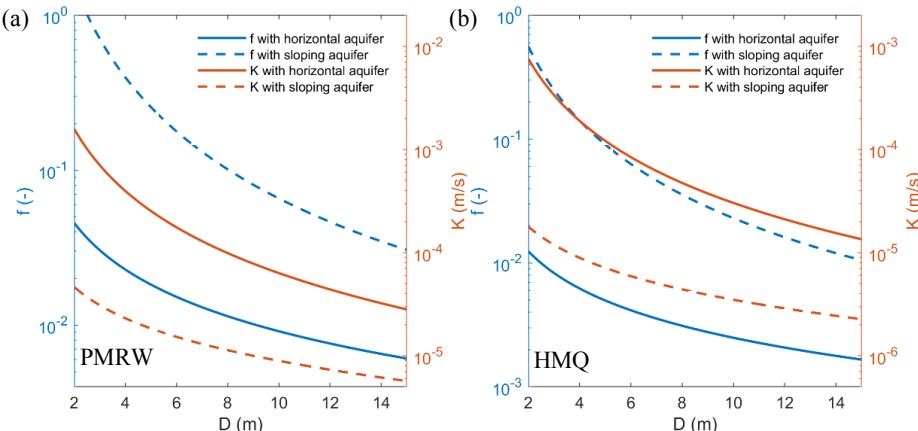

**Figure 6.** Comparison of the estimated $K$ and $f$ from sloping and horizontal aquifer with changing $D$ in (a) PMRW and (b) HMQ.