# Peer review of "Combining analytical solutions of Boussinesq equation with the modified Kozeny-Carman equation for estimation of catchment-scale hydrogeological parameters Man Gao1,2, Xi Chen1,2\*, and Jintao Liu3,4 1 Institute of Surface-Earth System Science, Tianjin University, Tianjin, China 5 2 Tianjin Key Laboratory of Earth Critical Zone Science and Sustainable Development in Bohai Rim, Tianjin University, "

_Hydrology and Earth System Sciences, 2019_

## Referee Comment (RC1) · Anonymous Referee #1 · 7 Oct 2019

Title: Combining analytical solutions of Boussinesq equation with the modified Kozeny–Carman equation for estimation of catchment-scale hydrogeological parameters

MS No.: hess-2019-453

GENERAL COMMENTS

This paper presents a means to constrain values of hydraulic conductivity, effective porosity, and aquifer depth estimated from recession analysis by using an empirical relationship between hydraulic conductivity and drainable porosity that is a function of

the pore size distribution, the latter being estimated from soil texture properties.

I think the concept has merit, is worth pursuing, and could lead to improved estimations of aquifer parameters. The theoretical part of the paper could be published but requires a better explanation of the assumptions that underlie the analytical solutions and a discussion of the implications of these assumptions when applying the parameter estimation technique to real world situations.

Where the paper falls short, and why I recommend this paper not be published, is in the application to real data. To claim that an early-time with $b = 3$ and late time regime with $b = 1$ exist in the aggregate of the data from any of the four watersheds is a very large stretch. An examination of individual recessions in the $dQ/dt$ vs $Q$ would likely reveal that such behavior ($b = 3$ to $b = 1$) is simply not present in the receding limbs of the hydrograph. Studies examining individual recessions have begun to show that the pattern in aggregated data, including apparent lower envelopes, does not represent constant aquifer properties (e.g., Biswal and Marani, 2010; Shaw and Riha, 2012; Mutzner et al., 2013; McMillan et al., 2014; Basso et al. 2015; Karlsen et al,. 2019; Santos et al., 2019). Jachens et al. (2019) in particular demonstrate the fallacy that the apparent pattern of the aggregated data in $dQ/dt$ vs $Q$ space (e.g., envelopes of $b = 3$ to 1 or other values of b estimated directly from aggregate) represent aquifer properties but rather arise from properties of the climate (i.e., magnitude and interarrival times of recharge events). Even to the extent the patterns do reflect aquifer properties, the authors do not show that the single Boussinesq aquifer (much less the simplifying assumptions required to achieve the analytical solutions) is a "good enough" representation of complex watershed made of multiple hillslopes and landscape scale heterogeneity in hydraulic properties to allow them to estimate aquifer properties using the proposed technique.

A more appropriate first test of the technique would be in a laboratory setting where the aquifer properties are consistent with a "Boussinesq" aquifer and the aquifer boundary conditions and initial conditions conform to those for which the analytical solutions were

derived. Another appropriate test could be for a single and well-instrumented hillslope where the boundary conditions and initial conditions can at least be measured, if not controlled. Datasets exists for both situations.

SPECIFIC COMMENTS

53: Assumptions of the analytical solutions of Brutsaert and Nieber (1977) do not precisely include a relatively humid setting. Precisely, the solutions assume an initial saturated thickness of equal depth along the length of the hillslope. Mathematically, this could result from a spatially uniform pulse recharge to an initially dry aquifer.

58-59: This statement is incorrect. Mendoza et al (2003) did not improve the solution of Parlange et al. (2001). They simply tweaked with the lower envelope fitting technique because they didn't observe a b = 3 regime.

87-89: How appropriate are these pedostransfer functions for fractured bedrock and less-weathered saprolite?

119: "x" is not the distance from the river to the hillslope ridge. This would be "B".

125-134: The authors leave out mentioning that linearization of Eq (2) is required to obtain the solutions presented here, and do not say what that linearization implies physically.

131: Eq. (3) is true as time goes to zero, or if advection is excluded, as Brutsaert (1994) states. The authors don't mention this.

136: I don't see Eq. (4) in Brutsaert (1994). Can the authors say what equation in Brutsaert (1994) this is meant to be?

153: How do the authors reconcile that fact that the b = 1 here is an artifact of the linearization of the sloping Boussinesq equation and does not occur if the equation is not linearized (e.g., Bogaart et al. 2013)?

162-169: The authors may want to see Roques et al. (2019), who present an improvement to Rupp and Selker (2006a).

213-214: What does these slopes, along with the aquifer depth and length, imply for the validity of the sloping and non-sloping Boussinesq and their various solutions? The "Hi" term is useful dimensionless number in this regard. See, for example, Brutsaert (2005) and Rupp and Selker (2006b).

226: Vertical vs horizontal K can be very different in bedrock. A falling head permeameter and the assumption about the shape of the wetting front made in estimating K make it not a suitable instrument for determining this. The authors should comment on possibly large errors in estimating K.

248-249 and Table 2: Are these values of drainable porosity high for bedrock and some saprolite? Can the authors compare with directly measured "f" for bedrock from other locations, to see if these values derived from the pedotransfer functions are unusually high?

252-253 and Figure 3: The recessions in Figure 3 do not support convergence to a value of 1 at late time. How do the authors reconcile the lack of data to support a lower envelope with a slope of 1with their subsequent estimation of the aquifer parameters? Clark et al. (2009) and Wang (2011) propose different conceptual models for PMRW, with at least two water sources contributing to the streamflow. Both are able to mimic the observed dQ/dt vs dQ pattern much better than can the single homogeneous Boussinesq aquifer assumed by the authors.

267-268: I would say that 35.6 days at PMRW is not relatively fast compared to 45 +/- 15 days, but well within that range. PMRW has a similar D to HMQ and WS10, so D does not explain why PMRW is "slower" than HMQ and WS10. Yet the authors use D to try to explain why SPG is

356-357: Can the authors comment on how these values for f compare to those in Brutsaert and Nieber (1977) and Brutseart and Lopez (1998)? They also calculated

values for f.

385-499: It is good that the authors consider to what degree discrepancies are due to riparian area impacts. Can the authors estimate the volume of water that must be stored to explain such discrepancies. Does it exceed riparian storage?

TECHNICAL/EDITORIAL CORRECTIONS

84: . . .of a soil pore. Or maybe better it is more correct to say the distribution of hydraulic radii of soil pores.

118: . . ., eta is the water table height above an impermeable layer,. . .

171-172: ". . .only the latest data are involved in the calculations. . ." It is not clear what this means.

200: The gamma term appears as an "r" in Eq. (19) in my pdf file.

219: Referring to any catchment as "famous" in this paper is unnecessary.

233: textures should be texture.

254-255: Should be either Brutsaert and Nieber (1977) or Brutsaert and Lopez (1998)

256 and 257: envelop should be envelope in both instances.

REFERENCES:

Bogaart, P. W., Rupp, D. E., Selker, J. S., & Van Der Velde, Y. (2013). Late‐-time drainage from a sloping Boussinesq aquifer. Water Resources Research, 49(11), 7498-7507.

Basso, S., Schirmer, M. and Botter, G.: On the emergence of heavy-tailed streamflow distributions, Adv. Water Resour., 82, 98–105, doi:10.1016/j.advwatres.2015.04.013, 2015.

Biswal, B. and Marani, M.: Geomorphological origin of recession curves, Geophys.

Res. Lett., 37(24), 1–5, doi:10.1029/2010GL045415, 2010.

Brutsaert, W. (1994). The unit response of groundwater outflow from a hillslope. Water Resources Research, 30(10), 2759-2763.

Brutsaert, W., & Lopez, J. P. (1998). Basin‐scale geohydrologic drought flow features of riparian aquifers in the southern Great Plains. Water Resources Research, 34(2), 233-240.

Brutsaert, W., & Nieber, J. L. (1977). Regionalized drought flow hydrographs from a mature glaciated plateau. Water Resources Research, 13(3), 637-643.

Brutsaert, W. (2005), Hydrology: An Introduction, Cambridge Univ. Press, New York.

Clark, M. P., Rupp, D. E., Woods, R. A., Tromp-van Meerveld, H. J., Peters, N. E., & Freer, J. E. (2009). Consistency between hydrological models and field observations: linking processes at the hillslope scale to hydrological responses at the watershed scale. Hydrological Processes: An International Journal, 23(2), 311-319.

Jachens, E. R., Rupp, D. E., Roques, C., & Selker, J. S. Recession analysis 42 years later-work yet to be done, Hydrology and Earth System Sciences Discussions, https://doi.org/10.5194/hess-2019-205, 2019.

Karlsen, R. H., Bishop, K., Grabs, T., Ottosson-Löfvenius, M., Laudon, H. and Seibert, J.: The role of landscape properties, storage and evapotranspiration on variability in streamflow recessions in a boreal catchment, J. Hydrol., (2019), doi:10.1016/j.jhydrol.2018.12.065, 2018.

Mcmillan, H., Gueguen, M., Grimon, E., Woods, R., Clark, M. and Rupp, D. E.: Spatial variability of hydrological processes and model structure diagnostics in a 50km2catchment, Hydrol. Process., 28(18), 4896–4913, doi:10.1002/hyp.9988, 2014.

Mendoza, G. F., Steenhuis, T. S., Walter, M. T., & Parlange, J. Y. (2003). Estimating basin-wide hydraulic parameters of a semi-arid mountainous watershed by recessionflow analysis. Journal of Hydrology, 279(1-4), 57-69.

Mutzner, R., Bertuzzo, E., Tarolli, P., Weijs, S. V., Nicotina, L., Ceola, S., Tomasic, N., Rodriguez-Iturbe, I., Parlange, M. B. and Rinaldo, A.: Geomorphic signatures on Brutsaert base flow recession analysis, Water Resour. Res., 49(9), 5462–5472, doi:10.1002/wrcr.20417, 2013.

Roques, C., Rupp, D. E., & Selker, J. S. (2017). Improved streamflow recession parameter estimation with attention to calculation of $-$ dQ/dt. Advances in water resources, 108, 29-43.

Rupp, D. E., & Selker, J. S. (2006a). Information, artifacts, and noise in dQ/dt$-$ Q recession analysis. Advances in water resources, 29(2), 154-160.

Rupp, D. E., & Selker, J. S. (2006b). On the use of the Boussinesq equation for interpreting recession hydrographs from sloping aquifers. Water Resources Research, 42(12).

Santos, A. C., Portela, M. M., Rinaldo, A. and Schaefli, B.: Estimation of streamflow recession parameters: New insights from an analytic streamflow distribution model, Hydrol. Process., doi:10.1002/hyp.13425, 2019.

Shaw, S. B. and Riha, S. J.: Examining individual recession events instead of a data cloud: Using a modified interpretation of dQ/dt-Q streamflow recession in glaciated watersheds to better inform models of low flow, J. Hydrol., 434–435, 46–54, doi:10.1016/j.jhydrol.2012.02.034, 2012.

Wang, D. (2011). On the base flow recession at the Panola mountain research watershed, Georgia, United States. Water

---

## Referee Comment (RC2) · Anonymous Referee #2 · 23 Oct 2019

Title: Combining analytical solution of Boussinesq equation with the modified Kozeny-Carman equation for estimation of catchment-scale hydrogeological parameters Ref. MS #HESS-2019-453

Overview:

The authors claim to develop a novel methodology to estimate the catchment-scale hydrogeological parameters of saturated hydraulic conductivity, K; drainable porosity, f; and the soil depth, D by combining the existing analytical solution of the Boussinesq

equation and the Kozeny-Carman equation. Subsequently, the developed approach is tested in four real-world study sites to conclude that the obtained soil parameters are well within the acceptable range. Although solutions to both the Boussinesq and the Kozeny-Carman equations exist in the literature, the authors' idea to combine both the solutions for estimating of aquifer property seems novel and interesting. It is worth mentioning that in earlier attempts to model the low flows from the delayed hillslope discharge, the soil depth, D is considered as a calibration parameter apart from K and f (e.g., Matonse and Kroll, 2009). The theoretical advancement, when established, could be helpful for modelling of the hydro-geologically ungauged basins wherein only streamflow data is available. However, there are several issues for which I am negative in recommending the paper for acceptance. Looking at the merits of the theoretical approach, the authors may be asked for a fully revised manuscript for resubmission.

Specific comments:

1. Getting first-hand information on spatial distribution of soil depth, D is easy in comparison to K and f. Unless the catchment under study is strictly ungauged and inaccessible, it can be obtained from the available well-logs directly and by vertical electrical sounding experiment indirectly that is neither costly nor time consuming. How would the authors justify the necessity of estimating the soil depth by analytical or empirical methods? This needs to be clearly justified in Introduction.

2. The title and the spatial scale of catchment chosen seem to be contradictory. Although, the authors claim for catchment-scale estimation, the study areas chosen do not reflect the same as all the four areas have the extent of 0.102 – 1.35 km2, which are only at the hillslope-scale. It is also reflected in the results obtained (Line #265-270), where the author state that the late-time recession is relatively fast except for SchÓğneben rock glacier (SPG) catchment. It could be due to the fact that a small hillslope would recede fast. The authors need to rethink and either change the title or test the approach at a suitable scale.

3. The delayed recession from the SPG could have resulted due to delayed release from snow and glacier melt. If this is the case, choosing this area poses a serious question as the Boussinesq equation and its solution deals with the draining hillslope aquifers, and not the glaciers. The authors can refer Winkler et al. (2016) for more details on the SPG.

4. It is good to see that the authors have considered both the early-time and late-time recessions; however, plotting at least one season of discharge data for each catchment would be more informative.

5. Form Fig. 3, it seems that the early-time and late-time recessions cannot be inferred from the analyzed recession data. Hence, a longer time series need to be analyzed with clear recession events (e.g., Rupp et al., 2009). As mentioned in Lines #257-258 and Fig. 3, it is not clear what is the physical basis of choosing the lower envelop lines with b=1 and b=3 to derive the recession intercepts. The range of this value looks too high.

6. The sensitivity analysis is not sufficient with only 10% change of independent variable (Fig. 5). Moreover, analysis for at least one more site would be informative.

7. Following Rupp and Selker (2006), consideration of variable time interval for recession analysis is interesting. The authors should mention the range of time interval considered for arriving at Fig. 3. Further, is this range same for all the four catchment or different?

8. Estimation of the hydraulic parameters considering both early- and late- time recession does not represent the same zone of aquifer that contribute to recession flow. Hence, it would not be better to say effective K, effective f rather than K and f.

9. Page 14: The field application results show that there is a huge gap between the estimated and observed soil hydraulic parameters, which may result in significant uncertainty in estimating the subsurface flux. Therefore, it is always advisable to calibrate

the parameters for their field use. An uncertainty analysis could strengthen the outcome of these results.

10. Eqs. (23)-(26) and Fig. 6: These are the ideal aquifer cases where K and f decrease with increase in the aquifer thickness, D. Therefore, these Eqs. could be far from the real hillslope cases.

11. The ability of the present approach in estimating the hydrogeological parameters can be tested fully by modelling the streamflow with the Boussinesq equation-based models with the estimated parameters instead of calibrating the model. I hope this would be the authors' next plan. However, rather than comparing the estimated parameters with the pedo-transfer function-based estimated results, testing the parameters in real modelling case would strengthen the claim in discussion.

Editorial:

1. Some long sentences need to be fragmented for clearer meaning: Lines #236-238, #242-243, #306-308

2. Line #115: Should be '... one-dimensional subsurface flow from the sloping aquifer' not 'on the'.

3. Reference for Eq. 1?

4. Line #119: From Fig. 1, the distance from river to ridge is B.

5. References for Eqs. 3 and 4?

6. Line #157: The verb form of the term should be 'recede', not 'recess'. Change accordingly at all subsequent appearances.

7. Lines: #157-160: It will be better to mention that the time duration between rainfall excess and beginning of recession depends upon catchment characteristic, extent, topography and depression storage.

8. Line #163: Replace '. . .in terms of. . .' by '. . .as per. . .'.

9. Line #200: Typo in Eq. 19. Replace 'r' by gamma.

10. Line #214: Change to '. . .fall under. . .'.

11. Line #221: Change to '. . .For SPG, the data published. . .'

12. Lines #234-235: Change to '. . .saturated hydraulic conductivity, K and soil/saprolite thickness, D. . . '. Change likewise at all other places.

13. Line #396: Change to 'Thus, detailed. . ., '.

14. Line #410: Change to '. . .equivalent values at. . .'.

References

Matonse, A.H., Kroll, C., 2009. Simulating low streamflows with hillslope storage models. Water Resour. Res. 45, 1–13. doi:10.1029/2007WR006529

Rupp, D.E., Schmidt, J., Woods, R.A., Bidwell, V.J., 2009. Analytical assessment and parameter estimation of a low-dimensional groundwater model. J. Hydrol. 377, 143–154. doi:10.1016/j.jhydrol.2009.08.018

Rupp, D.E., Selker, J.S., 2006. Information, artifacts, and noise in dQ/dt − Q recession analysis. Adv. Water Resour. 29, 154–160. doi:10.1016/J.ADVWATRES.2005.03.019

Winkler, G., Wagner, T., Pauritsch, M., Birk, S., Kellerer-Pirklbauer, A., Benischke, R., Leis, A., Morawetz, R., Schreilechner, M.G., Hergarten, S., 2016. Identification and assessment of groundwater flow and storage components of the relict Schöneben Rock Glacier, Niedere Tauern Range, Eastern Alps (Austria). Hydrogeol. J. 24, 937–953. doi:10.1007/s10040-015-1348-9

---

## Author Comment (AC1) · 9 Dec 2019

**Reply to the comments from the reviewer #1:**

**Anonymous Referee #1**

Title: Combining analytical solutions of Boussinesq equation with the modified Kozeny–Carman equation for estimation of catchment-scale hydrogeological parameters
MS No.: hess-2019-453

**GENERAL COMMENTS**

This paper presents a means to constrain values of hydraulic conductivity, effective porosity, and aquifer depth estimated from recession analysis by using an empirical relationship between hydraulic conductivity and drainable porosity that is a function of the pore size distribution, the latter being estimated from soil texture properties.

I think the concept has merit, is worth pursuing, and could lead to improved estimations of aquifer parameters. The theoretical part of the paper could be published but requires a better explanation of the assumptions that underlie the analytical solutions and a discussion of the implications of these assumptions when applying the parameter estimation technique to real world situations.

Where the paper falls short, and why I recommend this paper not be published, is in the application to real data. To claim that an early-time with b = 3 and late time regime with b = 1 exist in the aggregate of the data from any of the four watersheds is a very large stretch. An examination of individual recessions in the dQ/dt vs Q would likely reveal that such behavior (b = 3 to b = 1) is simply not present in the receding limbs of the hydrograph. Studies examining individual recessions have begun to show that the pattern in aggregated data, including apparent lower envelopes, does not represent constant aquifer properties (e.g., Biswal and Marani, 2010; Shaw and Riha, 2012; Mutzner et al., 2013; McMillan et al., 2014; Basso et al. 2015; Karlsen et al,. 2019; Santos et al., 2019). Jachens et al. (2019) in particular demonstrate the fallacy that the apparent pattern of the aggregated data in dQ/dt vs Q space (e.g., envelopes of b = 3 to 1 or other values of b estimated directly from aggregate) represent aquifer properties but rather arise from properties of the climate (i.e., magnitude and interarrival times of recharge events).

Response: We have revised the manuscript according to your suggestions. We described the assumptions of the analytical solutions derived from the Boussinesq equation in details in the revised manuscript.

We agree that the slopes of $-\mathrm{d}Q/\mathrm{d}t \sim Q$ from individual recessions could deviate the slopes of $b$=3 and 1 for the early-time and late-time recessions, respectively. According to reviewer's suggestions, we further analyzed the individual recessions of $Q(t) \sim t$ in semi-logarithm space as a complement. Individual recessions lasting at least 6 days after rainfall ceases are selected and the first two days data are removed to exclude the influence of surface flow.

In the analyzed catchments (PMRW and WS10), the individual events are selected for separating the early-time and the late-time recessions (Fig 1). The early-time recessions of individual segments can be analyzed by the following nonlinear equation (Brutsaert and Nieber, 1977; Tallaksen, 1995):

$$Q(t) = \left[Q(0)^{1-b_f} - a_f(1-b_f)t\right]^{\frac{1}{1-b_f}} \qquad (1)$$

where $Q$ is the discharge, $Q(0)$ is the initial discharge prior recession at $t=0$. This equation is equivalent to $-dQ(t)/dt = a_f Q^{b_f}$ ($b_f \neq 1$). Fig. 1 indicates that the parameter $a_f$ depends on initial discharge $Q(0)$. The parameters $a_f$ and $b_f$ are estimated by fitting the early recession segments in the first four days in this study. To meet the condition of $b_f=3$ for Eq. (15) in the previous manuscript, we selected the early recession segments that the slopes approach to 3, and the corresponding $a_f$ are listed in Table 1.

The tails of the late-time recessions of individual segments of $Q(t) \sim t$ in semi-logarithm space concentrate to a line (the master recession curve) in Fig. 1 (a, c). It indicates $b_s=1$ for the equation $-dQ(t)/dt = a_s Q^{b_s}$. The lower envelope with a slope of 1 is proved by Wang (2011) for the low discharges at the PMRW. Fitting the line of the master recession with slope $b_s=1$, we obtained the intercept of the line $a_s$ in Table 1.

Table 1 Properties of individual recession segments in two catchments

| Catchments | Numbers of recessions | Initial discharge (mm/d) | $a_s$ (s-1) | $a_f$ (m-6s) | $R_2$ |
|---|---|---|---|---|---|
| PMRW | 48 | 0.67~3.43 | $2.33 \times 10^{-7}$ | $1.5 \times 10^{-2} \sim 3.2 \times 10^{-1}$ $(8.0 \times 10^{-2})$ | 0.995 |
| WS10 | 53 | 0.79~6.93 | $4.34 \times 10^{-7}$ | $8.48 \times 10^{-2} \sim 6.97$ $(0.29)$ | 0.990 |

Comments: the value in the bracket refers to the mean value. $R_2$ is the mean coefficient of determination for all the fitted recessions.

(a)

[Figure]

(b)

[Figure]

[Figure]

Figure 1. Individual recessions of (a, c) $Q{\sim}t$ and (b, d) $-dQ/dt{\sim}Q$ for (a, b) PMRW and (c, d) WS10 (Recessions with different initial discharges are presented in different colors)

Then according to the analytical solution of one-dimensional subsurface flow from the sloping aquifer (Brutsaert, 2005), $K$ and $D$ can be obtained from implicit equations as follows (refer to the derivations in the previous manuscript):

$$4D^{\frac{-2+4\beta}{1+\beta}} + C_{s2}D^{\frac{-4+2\beta}{1+\beta}} = (a_sC_{s1}^{-1})(a_fC_f)^{\frac{1-\beta}{1+\beta}} \tag{2}$$

$$4K^{\frac{2-4\beta}{3}} + (a_f^{\frac{2}{3}}C_f^{\frac{2}{3}})C_{s2}K^{\frac{4-2\beta}{3}} = (a_sC_{s1}^{-1})(a_fC_f)^{\frac{1}{3}} \tag{3}$$

where $C_f = 8p/\pi\cos\alpha L^2\gamma^{-\beta}$, $\beta = 1/(3-\lambda)$, $C_{s1} = B^{-2}\gamma^{\beta}\pi^2 p\cos\alpha/4$, $C_{s2} = B^2\tan^2\alpha/(\pi^2 p^2)$, $\alpha$ is slope, $L$ is river length, $B$ is aquifer length, $p = 0.3465$, $\gamma$ and $\lambda$ are the parameters in pedotransfer function. Combing the modified Kozeny–Carman equation relates $K$ to $f$

$$f = \gamma^{-\beta}K^{\beta} \tag{4}$$

The catchment-scale hydrogeological parameters ($K$, $f$, and $D$) can be estimated simultaneously for each of the individual recessions.

For PMRW, the estimated $K$ values from various recessions are in the range of the field measurements (Fig. 2(a)). The estimated median value of $K$ from the individual recessions in WS10 is close to that from soil texture but is much smaller than the measured values (Fig. 2(a)). This could be attributed to the fact that the measurements were only taken at the upper soils (1.5 m in maximum) with abundant macropores (Harr, 1977), while the baseflow occurred at the underlying saprolite (McGuire and McDonnell, 2010) where $K$ is much small, such as $5 \times 10^{-6}$ m/s for the saprolite at PMRW (White et al., 2002).

Similarly, the estimated $D$ is mostly within the range of the measurements of the soil thickness for PMRW catchment. The range of the estimated $D$ are reasonable since the estimated $D$ represents an active thickness of water table variations in the deposits while the measured $D$ represents the entire thickness of deposits. The estimated median value of $D$ from individual recessions is close to the measured soil thickness in WS10. The estimated $f$ from soil texture approaches the maximum value of the estimated $f$ from the individual recessions in PMRW and WS10.

Besides, the estimated hydrogeological parameters of $K$ and $f$ increase with $Q(0)$ (Fig. (3)).

It indicates that the permeability and effective storage decrease with depths. Thus, the hydrogeological parameters analyzed from individual recessions reflect effect of vertical heterogeneity on baseflow recessions.

[Figure]

Figure 2. The estimated hydrogeological parameters of (a) *K*, (b) *D*, and (c) *f* from the individual recessions compared to the field measurements and the estimated values from soil texture. Note: the upper, middle, and lower circles in blue color represent the maximum, median, and minimum values of the estimates from individual recessions.

[Figure]

Figure 3. The relationships between initial discharge (*Q*(0)) and estimated effective parameters *K* and *f* for (a) PMRW and (b) WS10

Even to the extent the patterns do reflect aquifer properties, the authors do not show that the single Boussinesq aquifer (much less the simplifying assumptions required to achieve the analytical solutions) is a "good enough" representation of complex watershed made of multiple hillslopes and landscape scale heterogeneity in hydraulic properties to allow them to estimate aquifer properties using the proposed technique.

A more appropriate first test of the technique would be in a laboratory setting where the aquifer properties are consistent with a "Boussinesq" aquifer and the aquifer boundary conditions and initial conditions conform to those for which the analytical solutions were derived. Another appropriate test could be for a single and well-instrumented hillslope where the boundary conditions and initial conditions can at least be measured, if not controlled. Datasets exists for both situations.

Response: We agree that catchment flow comes from multiple hillslope flows and thus the landscape scale heterogeneity in hydrogeological properties could make difficulty to estimate

aquifer properties using the analytical solutions of the Boussinesq equation. Ideally, the numerical models that calibrated against the observation water tables and discharges will offer reliable estimations of the hydrogeological parameters in a heterogeneous catchment. However, such observations are sparse in the mountainous areas. In our selected catchments, the catchments are located in the headwater catchments and areas are small (i.e. less than 1.5 km2), which could reduce the spatial heterogeneity of landscapes and composition of multiple hillslopes on the baseflow analysis. WS10 has proven to represent a catchment dominated by hillslopes with negligible storage of water in riparian sediments (Harr, 1977; Triska et al., 1984).

The analytical solutions proposed by Brutsaert (1994) have been tested in the experiment sites and by the numerical simulations (Pauritsch et al., 2015; Rupp and Selker, 2006). Although analytical solutions of Boussinesq equation are derived from the simplified aquifer, they have been widely used to analyze baseflow characteristics and estimate catchment parameters (Brutsaert and Hiyama, 2012; Lyon et al., 2009; Pacheco and Van der Weijden, 2012; Sánchez-Murillo, et al., 2015; Thomas et al., 2015; Vannier et al., 2014; Zhang et al., 2009).

In our analysis, we selected the experimental catchments with relatively detail information of catchment properties, such as the measured hydraulic conductivities, soil thickness and porosity, in order to valid the estimated parameters from our derived equations.

**SPECIFIC COMMENTS**

53: Assumptions of the analytical solutions of Brutsaert and Nieber (1977) do not precisely include a relatively humid setting. Precisely, the solutions assume an initial saturated thickness of equal depth along the length of the hillslope. Mathematically, this could result from a spatially uniform pulse recharge to an initially dry aquifer.

Response: We agree with the reviewer's opinion that the analytical solutions do not restrict to humid setting. We expressed assumptions of the derived solutions in more details in the revised manuscript.

58-59: This statement is incorrect. Mendoza et al (2003) did not improve the solution of Parlange et al. (2001). They simply tweaked with the lower envelope fitting technique because they didn't observe a b = 3 regime.

Response: We revised this sentence in the manuscript.

87-89: How appropriate are these pedotransfer functions for fractured bedrock and less-weathered saprolite?

Response: The pedotransfer functions are derived from soils. In this study, the estimated parameters derived by the pedotransfer functions are regarded as the values of the hydrogeological parameters in the porous saprolite equivalent to a specific soil. For example, in PMRW and W10, $K$ estimated from the soil texture is equivalent to $K$ in silt loam and clay loam, respectively. We discussed this aspect in the revised manuscript.

119: "x" is not the distance from the river to the hillslope ridge. This would be "B".

Response: Here, $x$ represents the distance from river to a specific position on the hillslope. We revised this expression in the manuscript.

125-134: The authors leave out mentioning that linearization of Eq (2) is required to obtain the solutions presented here, and do not say what that linearization implies physically.
Response: This expression has been revised according to your suggestions.

131: Eq. (3) is true as time goes to zero, or if advection is excluded, as Brutsaert (1994) states. The authors don't mention this.
Response: We added this condition in the statement of analytical solution from Brutsaert (1994).

136: I don't see Eq. (4) in Brutsaert (1994). Can the authors say what equation in Brutsaert (1994) this is meant to be?
Response: The analytical solution in Eq. (4) refers to the first term of Eq. (17) proposed by Brutsaert (1994). We revised the expression.

153: How do the authors reconcile that fact that the b = 1 here is an artifact of the linearization of the sloping Boussinesq equation and does not occur if the equation is not linearized (e.g., Bogaart et al. 2013)?
Response: Yes, the b = 1 could be an artifact of the linearization of the sloping Boussinesq equation. However, Pauritsch et al. (2015) found that it is more convenient to use the analytical solution from Brutsaert (1994) to estimate hydrogeological parameters, particularly at slope angles greater than 10°, by comparing different analytical solutions with numerical solution. We adopted this statement.

162-169: The authors may want to see Roques et al. (2017), who present an improvement to Rupp and Selker (2006a).
Response: We adopted the improvement method from Roques et al. (2017) to generate more reliable recession data points in the revised manuscript.

213-214: What does these slopes, along with the aquifer depth and length, imply for the validity of the sloping and non-sloping Boussinesq and their various solutions? The "Hi" term is useful dimensionless number in this regard. See, for example, Brutsaert (2005) and Rupp and Selker (2006b).
Response: As it is shown in Eqs. (4) and (9) in the previous manuscript, the dimensionless parameter Hi, represents the relative magnitude of the slope term, i.e. the effect of gravity, versus the diffusion term. If Hi/2 greater than $\pi$ (tending to sloping), the gravity term is dominant, otherwise the diffusion term is the dominant one (tending to non-sloping).

226: Vertical vs horizontal K can be very different in bedrock. A falling head permeameter and the assumption about the shape of the wetting front made in estimating K make it not a suitable instrument for determining this. The authors should comment on possibly large errors in estimating K.
Response: We added sentence on possible errors between our estimates and the measurements in the section of discussions.

248-249 and Table 2: Are these values of drainable porosity high for bedrock and some saprolite?

Can the authors compare with directly measured "f" for bedrock from other locations, to see if these values derived from the pedotransfer functions are unusually high?

Response: The drainable porosity of saprolite and shallow weathered bedrock can be large or even approximate that of soil. For example, Hubbert et al. (2001) showed that the drainable porosity of weathered granitic bedrock at 0.8~1.6 m can be larger than 0.2 (Fig. 2 in the his published paper); according to measurements by Graham et al., (1997), the porosity of macro-void for weathered granitic bedrock is greater than 0.1 mm in upland areas of California, leading to the effective porosity in a range of 0.089~0.149 at entisol site and 0.068~0.115 at alfisol site.
  We added these comparisons in the revised manuscript.

252-253 and Figure 3: The recessions in Figure 3 do not support convergence to a value of 1 at late time. How do the authors reconcile the lack of data to support a lower envelope with a slope of 1 with their subsequent estimation of the aquifer parameters? Clark et al. (2009) and Wang (2011) propose different conceptual models for PMRW, with at least two water sources contributing to the streamflow. Both are able to mimic the observed dQ/dt vs dQ pattern much better than can the single homogeneous Boussinesq aquifer assumed by the authors.

Response: Fig. 1 shows that individual recessions of $Q\sim t$ in semi-logarithm space approach to a line with b=1 in our study.
  In our re-analyzed results, the fitting of the early-time and late-time individual recessions obtained fast flow recession and slow flow recession, respectively, indicating that two water sources contribute to the streamflow.

267-268: I would say that 35.6 days at PMRW is not relatively fast compared to 45 +/-15 days, but well within that range. PMRW has a similar D to HMQ and WS10, so D does not explain why PMRW is "slower" than HMQ and WS10. Yet the authors use D to try to explain why SPG is

Response: These sentences have been revised as suggested.

356-357: Can the authors comment on how these values for f compare to those in Brutsaert and Nieber (1977) and Brutseart and Lopez (1998)? They also calculated values for f.

Response: The estimated values of $f$ from Brutsaert and Nieber (1977) and Brutsaert and Lopez (1998) are around 0.02 in terms of the geometric mean value, which are underestimated for soils. The reason may be their overestimations of $K$ based on horizontal aquifer assumption.

385-499: It is good that the authors consider to what degree discrepancies are due to riparian area impacts. Can the authors estimate the volume of water that must be stored to explain such discrepancies. Does it exceed riparian storage?

Response: We added this component in our revised manuscript. For example, at PMRW where the riparian, hillslope, and bedrock outcrop area consist the catchment area of 15%, 75%, and 10%, respectively. The soil thickness is about 1 m at hillslope and can reach 5 m in riparian area. When the riparian aquifer is the fully saturated, the water storage capacity for the riparian area can be calculated by multiplying the aquifer thickness (5 m) and drainable porosity (0.27). When assuming groundwater in the whole catchment only stored in riparian area, water storage of the whole catchment is 202.5 mm (0.15 × 5 m × 0.27).

The storage for each event in the catchment aquifer can be calculated from discharge as $S = \int_0^T Q(t)dt + Q(T)/a_s$, where $T$ is the transition time from the early-time recession to the late-time recession. As listed in Table 1, the largest initial discharge is 3.43 mm/day, and the corresponding water storage estimated in the whole catchment is 84.3 mm. Thus, riparian storage (202.5 mm) can satisfy the groundwater draining storage (84.3 mm).

**TECHNICAL/EDITORIAL CORRECTIONS**

84: …of a soil pore. Or maybe better it is more correct to say the distribution of hydraulic radii of soil pores.
Response: It has been revised as suggested.

118: …, eta is the water table height above an impermeable layer, …
Response: It has been revised as suggested.

171-172: "… only the latest data are involved in the calculations …" It is not clear what this means.
Response: We revised this sentence to make it clear.

200: The gamma term appears as an "r" in Eq. (19) in my pdf file.
Response: We revised this.

219: Referring to any catchment as "famous" in this paper is unnecessary.
Response: We revised this sentence.

233: textures should be texture.
Response: We corrected this word as suggested.

254-255: Should be either Brutsaert and Nieber (1977) or Brutsaert and Lopez (1998)
Response: It a mistake. We revised this.

256 and 257: envelop should be envelope in both instances.
Response: We corrected this word.

References:
Brutsaert, W.: The unit response of groundwater outflow from a hillslope, Water Resour. Res., 30(10), 2759-2763, 1994.
Brutsaert, W.: Hydrology: an introduction, Cambridge University Press, 2005.
Brutsaert, W., and Lopez, J. P.: Basin-scale geohydrologic drought flow features of riparian aquifers in the southern Great Plains, Water Resour. Res., 34(2), 233-240, 1998.
Brutsaert, W., and Nieber J. L.: Regionalized drought flow hydrographs from a mature glaciated plateau, Water Resour. Res., 13(3), 637-643, 1977.
Brutsaert, Wilfried, and Tetsuya Hiyama: The determination of permafrost thawing trends from long-term streamflow measurements with an application in eastern Siberia, J Geophys., Res.:

Atmos., 117.D22, 2012.

Graham, R. C., Anderson, M. A., Sternberg, P. D., Tice, K. R., and Schoeneberger, P. J.: Morphology, porosity, and hydraulic conductivity of weathered granitic bedrock and overlying soils, Soil Sci. Soc. Am. J., 61(2), 516-522, 1997.

Harr, R. D.: Water flux in soil and subsoil on a steep forested slope, J. Hydrol., 33(1-2), 37-58, 1977.

Hubbert, K.R., Graham, R.C. and Anderson, M.A.: Soil and weathered bedrock. Soil Sci. Soc. Am. J., 65(4), 1255-1262, 2001.

Jachens, E. R., Rupp, D. E., Roques, C., and Selker, J. S.: Recession analysis 42 years later – work yet to be done, Hydrol. Earth Syst. Sci. Discuss., https://doi.org/10.5194/hess-2019-205, in review, 2019.

Lyon, S. W., Destouni, G., Giesler, R., Humborg, C., Mörth, M., Seibert, J., Karlsson, J., and Troch, P. A.: Estimation of permafrost thawing rates in a sub-arctic catchment using recession flow analysis, Hydrol. Earth Syst. Sci, 13, 595-604, 2009.

McGuire, K. J., and McDonnell, J. J.: Hydrological connectivity of hillslopes and streams: Characteristic time scales and nonlinearities, Water Resour. Res., 46(10), 2010.

Pauritsch, M., Birk, S., Wagner, T., Hergarten, S., and Winkler, G.: Analytical approximations of discharge recessions for steeply sloping aquifers in alpine catchments, Water Resour. Res., 51(11), 8729-8740, 2015.

Pacheco, Fernando AL, and Cornelis H. Van der Weijden: Integrating topography, hydrology and rock structure in weathering rate models of spring watersheds, J. Hydrol., 428: 32-50, 2012.

Roques, C., Rupp, D.E. and Selker, J. S.: Improved streamflow recession parameter estimation with attention to calculation of −dQ/dt, Adv. Water Resour., 108, 29-43, 2017.

Rupp, D. E., and Selker, J. S.: On the use of the Boussinesq equation for interpreting recession hydrographs from sloping aquifers, Water Resour. Res., 42(12), 2006.

Sánchez-Murillo, R., Brooks, E. S., Elliot, W. J., Gazel, E., and Boll, J.: Baseflow recession analysis in the inland Pacific Northwest of the United States, Hydrogeol. J., 23(2), 287-303, 2015.

Tallaksen, L. M.: A review of baseflow recession analysis, J. Hydrol., 165(1-4), 349-370, 1995.

Thomas, B. F., Vogel, R. M., and Famiglietti, J. S.: Objective hydrograph baseflow recession analysis, J. Hydrol., 525, 102-112, 2015.

Triska, F. J., Sedell, J. R., Cromack Jr, K., Gregory, S. V., and McCorison, F. M.: Nitrogen budget for a small coniferous forest stream, Ecological Monographs, 54(1), 119-140, 1984.

Vannier, O., Braud, I., and Anquetin, S.: Regional estimation of catchment-scale soil properties by means of streamflow recession analysis for use in distributed hydrological models, Hydrol. Process., 28(26), 6276-6291, 2014.

Wang, D.: On the base flow recession at the Panola mountain research watershed, Georgia, United States, Water Resour. Res., 47(3), 2011.

White, A., Blum, A. E., Schulz, M. S., Huntington, T. G., Peters, N. E., Stonestrom, D. A.: Chemical weathering of the Panola Granite: Solute and regolith elemental fluxes and the weathering rate of biotite. In *Water-Rock Interactions, Ore Deposits, Environmental Geochemistry: A Tribute to David A. Crerar,* Hellmann R., Wood S. A. (Eds.). The Geochemical Society, Special Publication No. 7, 37-59, 2002.

Zhang, L., Chen, Y. D., Hickel, K., and Shao, Q.: Analysis of low-flow characteristics for catchments in Dongjiang Basin, China. Hydrogeol. J., 17(3), 631-640, 2009.

---

## Author Comment (AC2) · 9 Dec 2019

**Reply to the comments from the reviewer #2:**

**Anonymous Referee #2**

Title: Combining analytical solution of Boussinesq equation with the modified KozenyCarman equation for estimation of catchment-scale hydrogeological parameters
Ref. MS #HESS-2019-453

**Overview:**

The authors claim to develop a novel methodology to estimate the catchment-scale hydrogeological parameters of saturated hydraulic conductivity, K; drainable porosity, f; and the soil depth, D by combining the existing analytical solution of the Boussinesq equation and the Kozeny-Carman equation. Subsequently, the developed approach is tested in four real-world study sites to conclude that the obtained soil parameters are well within the acceptable range. Although solutions to both the Boussinesq and the Kozeny-Carman equations exist in the literature, the authors' idea to combine both the solutions for estimating of aquifer property seems novel and interesting. It is worth mentioning that in earlier attempts to model the low flows from the delayed hillslope discharge, the soil depth, D is considered as a calibration parameter apart from K and f (e.g., Matonse and Kroll, 2009). The theoretical advancement, when established, could be helpful for modelling of the hydro-geologically ungauged basins wherein only streamflow data is available. However, there are several issues for which I am negative in recommending the paper for acceptance. Looking at the merits of the theoretical approach, the authors may be asked for a fully revised manuscript for resubmission.

We are very grateful for your comments that help us to improve the quality of the manuscript. The whole paper has been thoroughly revised. The point-by-point responses are provided below.

**Specific comments:**

1. Getting first-hand information on spatial distribution of soil depth, D is easy in comparison to K and f. Unless the catchment under study is strictly ungauged and inaccessible, it can be obtained from the available well-logs directly and by vertical electrical sounding experiment indirectly that is neither costly nor time consuming. How would the authors justify the necessity of estimating the soil depth by analytical or empirical methods? This needs to be clearly justified in Introduction.

Response: We agree that the soil thickness can be obtained from the available well-logs and by geophysical surveys, but it is still time consuming for the details of the soil depth distributions in a catchment. The detailed measurements of the soil thickness are only available in a few experimental catchments. Even these measured thicknesses are available, they cannot be directly used to represent the "effective depth" of flow dynamic domain.

We described the necessity in the revised manuscript.

2. The title and the spatial scale of catchment chosen seem to be contradictory. Although, the authors claim for catchment-scale estimation, the study areas chosen do not reflect

the same as all the four areas have the extent of 0.102 – 1.35 km2, which are only at the hillslope-scale. It is also reflected in the results obtained (Line #265- 270), where the author state that the late-time recession is relatively fast except for Schöneben rock glacier (SPG) catchment. It could be due to the fact that a small hillslope would recede fast. The authors need to rethink and either change the title or test the approach at a suitable scale.

Response: We revised the title to focus on hydrogeological parameters in small headwater catchments. The four small catchments can be conceptualized as hillslopes. As to the fast recession at small catchments, it is attributed to the steep slope and limited plain riparian area.

3. The delayed recession from the SPG could have resulted due to delayed release from snow and glacier melt. If this is the case, choosing this area poses a serious question as the Boussinesq equation and its solution deals with the draining hillslope aquifers, and not the glaciers. The authors can refer Winkler et al. (2016) for more details on the SPG.

Response: According to Winkler et al. (2016), the SPG is a catchment with relict rock glaciers, which is a kind of rock glaciers and ice has disappeared. Thus, there is no influence from glaciers. In our study, the recession data were selected in winter when the meltwater from snow and recharge from other sources is limited as temperature is below zero.

4. It is good to see that the authors have considered both the early-time and late-time recessions; however, plotting at least one season of discharge data for each catchment would be more informative.

Response: We plotted each recession segment in $Q \sim t$ and $-\mathrm{d}Q/\mathrm{d}t \sim Q$ forms in different colors as shown in Fig. 1 below and calibrated the parameters of each recession in the revised manuscript (Table 1 and Figure 2 below).

5. Form Fig. 3, it seems that the early-time and late-time recessions cannot be inferred from the analyzed recession data. Hence, a longer time series need to be analyzed with clear recession events (e.g., Rupp et al., 2009). As mentioned in Lines #257-258 and Fig. 3, it is not clear what is the physical basis of choosing the lower envelop lines with b=1 and b=3 to derive the recession intercepts. The range of this value looks too high.

Response: We agree that it not visual to infer the early-time and late-time recessions for aggregated data points of $-\mathrm{d}Q/\mathrm{d}t \sim Q$ and a longer time series is helpful. Considering the reviewer's comments, we further analyzed the individual recessions of $Q(t) \sim t$ in semi-logarithm space as a complement and construct a master recession. Individual recessions lasting at least 6 days after rainfall ceases are selected and the first two days data are removed to exclude the influence of surface flow.

In the analyzed catchments (PMRW and WS10), the individual events are selected for separating the early-time and the late-time recessions (Fig 1). The early-time recessions of individual segments can be analyzed by the following nonlinear equation (Brutsaert and Nieber, 1977; Tallaksen, 1995):

$$Q(t) = \left[Q(0)^{1-b_f} - a_f(1-b_f)t\right]^{\frac{1}{1-b_f}} \qquad (1)$$

where $Q$ is the discharge, $Q(0)$ is the initial discharge prior recession at $t=0$. This equation is equivalent to $-dQ(t)/dt = a_f Q^{b_f}$. Fig. 1 indicates that the parameter $a_f$ depends on initial discharge $Q(0)$. The parameters $a_f$ and $b_f$ are estimated by fitting the early recession segments in the first four days in this study. To meet the condition of $b_f=3$ for Eq. (15) in the manuscript, we selected the early recession segments that the slopes approach to 3, and the corresponding $a_f$ are listed in Table 1.

The tails of the late-time recessions of individual segments of $Q(t) \sim t$ in semi-logarithm space concentrate to a line (the master recession curve) in Fig. 1 (a, c). It indicates $b_s=1$ for the equation $-dQ(t)/dt = a_s Q^{b_s}$. The lower envelope with a slope of 1 is proved by Wang (2011) for the low discharges at the PMRW. Fitting the line of the master recession with slope $b_s=1$, we obtained the intercept of the line $a_s$ in Table 1.

Table 1 Properties of individual recession segments in two catchments

| Catchments | Numbers of recessions | Initial discharge (mm/d) | $a_s$ (s-1) | $a_f$ (m-6s) | $R_2$ |
|---|---|---|---|---|---|
| PMRW | 48 | 0.67~3.43 | $2.33 \times 10^{-7}$ | $1.5 \times 10^{-2} \sim 3.2 \times 10^{-1}$ ($8.0 \times 10^{-2}$) | 0.995 |
| WS10 | 53 | 0.79~6.93 | $4.34 \times 10^{-7}$ | $8.48 \times 10^{-2} \sim 6.97$ (0.29) | 0.990 |

Comments: the value in the bracket refers to the mean value. $R_2$ is the mean coefficient of determination for all the fitted recessions.

[Figure]

[Figure]

[Figure]

Figure 1. Individual recessions of (a, c) $Q \sim t$ and (b, d) $-dQ/dt \sim Q$ for (a, b) PMRW and (c, d) WS10 (Recessions with different initial discharges are presented in different colors)

Then according to the analytical solution of one-dimensional subsurface flow from the sloping aquifer (Brutsaert, 2005), $K$ and $D$ can be obtained from implicit equations as follows (refer to the derivations in the previous manuscript):

$$4D^{\frac{-2+4\beta}{1+\beta}} + C_{s2}D^{\frac{-4+2\beta}{1+\beta}} = (a_s C_{s1}^{-1})(a_f C_f)^{\frac{1-\beta}{1+\beta}} \qquad (2)$$

$$4K^{\frac{2-4\beta}{3}} + (a_f^{\frac{2}{3}} C_f^{\frac{2}{3}})C_{s2}K^{\frac{4-2\beta}{3}} = (a_s C_{s1}^{-1})(a_f C_f)^{\frac{1}{3}} \qquad (3)$$

where $C_f = 8p/\pi \cos \alpha L^2 \gamma^{-\beta}$, $\beta = 1/(3-\lambda)$, $C_{s1} = B^{-2}\gamma^\beta \pi^2 p \cos \alpha/4$, $C_{s2} = B^2 \tan^2 \alpha /(\pi^2 p^2)$, $\alpha$ is slope, $L$ is river length, $B$ is aquifer length, $p = 0.3465$, $\gamma$ and $\lambda$ are the parameters in pedotransfer function. Combing the modified Kozeny–Carman equation relates $K$ to $f$

$$f = \gamma^{-\beta} K^\beta \qquad (4)$$

The catchment-scale hydrogeological parameters ($K$, $f$, and $D$) can be estimated simultaneously for each of the individual recessions.

For PMRW, the estimated $K$ values from various recessions are in the range of the field measurements (Fig. 2(a)). The estimated median value of $K$ from the individual recessions in WS10 is close to that from soil texture but is much smaller than the measured values (Fig. 2(a)). This could be attributed to the fact that the measurements were only taken at the upper soils (1.5 m in maximum) with abundant macropores (Harr, 1977), while the baseflow occurred at the underlying saprolite (McGuire and McDonnell, 2010) where $K$ is much small, such as $5 \times 10^{-6}$ m/s for the saprolite at PMRW (White et al., 2002).

Similarly, the estimated $D$ is mostly within the range of the measurements of the soil thickness for PMRW catchment. The range of the estimated $D$ are reasonable since the estimated $D$ represents an active thickness of water table variations in the deposits while the measured $D$ represents the entire thickness of deposits. The estimated median value of $D$ from individual recessions is close to the measured soil thickness in WS10. The

estimated *f* from soil texture approaches the maximum value of the estimated *f* from the individual recessions in PMRW and WS10.

Besides, the estimated hydrogeological parameters of *K* and *f* increase with $Q(0)$ (Fig. (3)). It indicates that the permeability and effective storage decrease with depths. Thus, the hydrogeological parameters analyzed from individual recessions reflect effect of vertical heterogeneity on baseflow recessions.

[Figure]

Figure 2. The estimated hydrogeological parameters of (a) *K*, (b) *D*, and (c) *f* from the individual recessions compared to the field measurements and the estimated values from soil texture. Note: the upper, middle, and lower circles in blue color represent the maximum, median, and minimum values of the estimates from individual recessions.

[Figure]

Figure 3. The relationships between initial discharge ($Q(0)$) and estimated effective parameters *K* and *f* for (a) PMRW and (b) WS10

6. The sensitivity analysis is not sufficient with only 10% change of independent variable (Fig. 5). Moreover, analysis for at least one more site would be informative.

Response: We did sensitivity analysis to all these four headwater catchments and conducted 10%, 20%, and 50% changes of each variable for WS10 to make the results more informative as shown in Fig. 4. We added more analyses in the section 4.3 in the revised manuscript.

Besides, these analytical solutions Eqs. (20) ~ (22) in the previous manuscript also show the importance of independent factors to these hydrogeological parameters. The sensitivity analysis makes the results visualization.

[Figure]

Figure 4. Sensitivity analysis of the estimated effective parameters ($K$, $D$, and $f$) in terms of the relative changes caused by changes of any variables by (a) 10% in SPG, (b) 10% in PMRW, (c) 10% in HMQ, (d) 10% in WS10, (e) 20% in WS10, and (f) 50% in WS10.

7. Following Rupp and Selker (2006), consideration of variable time interval for recession analysis is interesting. The authors should mention the range of time interval considered for arriving at Fig. 3. Further, is this range same for all the four catchment or different?

Response: The ranges of time interval are different for the four catchments. The range depends on the length of individual recessions. We revised the method to generate recession data points in -d$Q$/d$t$~$Q$ form. An improved method proposed by Roques et al. (2017) are used instead of Rupp and Selker (2006).

8. Estimation of the hydraulic parameters considering both early- and late- time recession does not represent the same zone of aquifer that contribute to recession flow. Hence, it would not be better to say effective K, effective f rather than K and f.

Response: We agree with the reviewer's suggestions, the names of effective $K$, effective $f$ rather than $K$ and $f$ are more suitable. We revised these in the manuscript as the reviewer's suggestions.

9. Page 14: The field application results show that there is a huge gap between the estimated and observed soil hydraulic parameters, which may result in significant uncertainty in estimating the subsurface flux. Therefore, it is always advisable to calibrate the parameters for their field use. An uncertainty analysis could strengthen the

outcome of these results.

Response: In the revised manuscript, we obtained the ranges of the estimated effective parameters from the analysis of individual recessions. The ranges of the estimated $D$ are narrower than the measured $D$. This is reasonable since the estimated effective $D$ represents an active thickness of water table variations in the deposits while the measured $D$ represents the entire thickness of deposits. Surely, there is still a gap between estimated and observed $K$. The significant uncertainty can come from the accuracy of observed streamflow, heterogeneity in catchment landscape, and the linearization of analytical solutions. Our estimated effective parameter could be viewed as a representative parameter set used as the initial ranges of these parameters in hydrological models. We expressed these in the revised manuscript.

We cannot directly analyze the uncertainty in terms of our method and data, but the sensitivity analysis and the comparison between the estimated and measured values partly reflect the uncertainty in our analysis. We discussed it in the revised manuscript.

10. Eqs. (23)-(26) and Fig. 6: These are the ideal aquifer cases where K and f decrease with increase in the aquifer thickness, D. Therefore, these Eqs. could be far from the real hillslope cases.

Response: In the cases of Eqs. (23)~(24) and (25)~(26) in the previous manuscript, two equations express the three hydrogeological parameters. So at least one variable such as $D$ is dependent on the other two, as shown in Fig 6. This figure shows relationships of $K$ and $f$ with increase in the aquifer thickness, $D$. It does not mean the spatial variations of $K$ and $f$ with $D$ in the real hillslope cases. Actually, $K$ and $f$ are independent of $D$ in our study when the third equation (the modified Kozeny–Carman equation) is introduced as shown in Eqs. (17) ~ (18) and (20) ~ (22) in the previous manuscript.

11. The ability of the present approach in estimating the hydrogeological parameters can be tested fully by modelling the streamflow with the Boussinesq equation-based models with the estimated parameters instead of calibrating the model. I hope this would be the authors' next plan. However, rather than comparing the estimated parameters with the pedo-transfer function-based estimated results, testing the parameters in real modelling case would strengthen the claim in discussion.

Response: We agree that it would be great important to test effectivity of these estimated parameters by modelling the streamflow. Actually, the estimated catchment-scale parameters based on recession analysis of the Boussinesq equation have been applied for hydrological modeling. For example, Vannier et al. (2016) estimated aquifer thickness and saturated hydraulic conductivity based on Brutsaert-Nieber method, which were applied for the hydrological model parameters and achieved a high simulation accuracy.

We discussed this aspect of application in hydrological modelling in the revised manuscript.

**Editorial:**
1. Some long sentences need to be fragmented for clearer meaning: Lines #236-238,

**242-243, #306-308**
Response: We revised these sentences as suggested.

2. Line #115: Should be '. . . one-dimensional subsurface flow from the sloping aquifer' not 'on the'.
Response: We revised the manuscript as suggested.

3. Reference for Eq. 1?
Response: We added a reference to the equation.

4. Line #119: From Fig. 1, the distance from river to ridge is B.
Response: We revised this sentence.

5. References for Eqs. 3 and 4?
Response: We added references to these equations.

6. Line #157: The verb form of the term should be 'recede', not 'recess'. Change accordingly at all subsequent appearances.
Response: We revised it as suggested.

7. Lines: #157-160: It will be better to mention that the time duration between rainfall excess and beginning of recession depends upon catchment characteristic, extent, topography and depression storage.
Response: We added a sentence to mention this content.

8. Line #163: Replace '. . .in terms of. . .' by '. . .as per. . .'.
Response: We revised it as suggested.

9. Line #200: Typo in Eq. 19. Replace 'r' by gamma.
Response: We revised it as suggested.

10. Line #214: Change to '. . .fall under. . .'.
Response: We revised it as suggested.

11. Line #221: Change to '. . .For SPG, the data published. . .'
Response: We revised it as suggested.

12. Lines #234-235: Change to '. . .saturated hydraulic conductivity, K and soil/saprolite thickness, D. . . '. Change likewise at all other places.
Response: We revised it as suggested.

13. Line #396: Change to 'Thus, detailed. . .,'. 14. Line #410: Change to '. . .equivalent values at. . .'.
Response: We revised it as suggested.

**References**

Brutsaert, W.: The unit response of groundwater outflow from a hillslope, Water Resour. Res., 30(10), 2759-2763, 1994.

Brutsaert, W.: Hydrology: an introduction, Cambridge University Press, 2005.

Brutsaert, W., and Nieber J. L.: Regionalized drought flow hydrographs from a mature glaciated plateau, Water Resour. Res., 13(3), 637-643, 1977.

Harr, R. D.: Water flux in soil and subsoil on a steep forested slope, J. Hydrol., 33(1-2), 37-58, 1977.

McGuire, K. J., and McDonnell, J. J.: Hydrological connectivity of hillslopes and streams: Characteristic time scales and nonlinearities, Water Resour. Res., 46(10), 2010.

Roques, C., Rupp, D.E. and Selker, J. S.: Improved streamflow recession parameter estimation with attention to calculation of -dQ/dt, Adv. Water Resour., 108, 29-43, 2017.

Rupp, D. E., and Selker, J. S.: Information, artifacts, and noise in -dQ/dt-Q recession analysis, Adv. Water Resour., 29(2), 154-160, 2006.

Tallaksen, L. M.: A review of baseflow recession analysis. J. Hydrol., 165(1-4), 349-370, 1995.

Triska, F. J., Sedell, J. R., Cromack Jr, K., Gregory, S. V., McCorison, F. M.: Nitrogen budget for a small coniferous forest stream. Ecological Monographs, 54(1), 119-140, 1984.

Vannier, O., Anquetin, S., and Braud, I.: Investigating the role of geology in the hydrological response of Mediterranean catchments prone to flash-floods: Regional modelling study and process understanding. Journal of Hydrology, 541, 158-172, 2016.

Wang, D.: On the base flow recession at the Panola mountain research watershed, Georgia, United States, Water Resour. Res., 47(3), 2011.

White, A., Blum, A. E., Schulz, M. S., Huntington, T. G., Peters, N. E., Stonestrom, D. A.: Chemical weathering of the Panola Granite: Solute and regolith elemental fluxes and the weathering rate of biotite. In *Water-Rock Interactions, Ore Deposits, Environmental Geochemistry: A Tribute to David A. Crerar,* Hellmann R., Wood S. A. (Eds.). The Geochemical Society, Special Publication No. 7, 37-59, 2002.

Winkler, G., Wagner, T., Pauritsch, M., Birk, S., Kellerer-Pirklbauer, A., Benischke, R., Leis, A., Morawetz, R., Schreilechner, M.G., Hergarten, S., 2016. Identification and assessment of groundwater flow and storage components of the relict Schöneben Rock Glacier, Niedere Tauern Range, Eastern Alps (Austria). Hydrogeol. J. 24, 937– 953. doi:10.1007/s10040-015-1348-9.